# Ezrin, radixin, and moesin are dispensable for macrophage migration and cellular cortex mechanics

Perrine Verdys [1,2], Javier Rey Barroso [1], Adeline Girel[1], Joseph Vermeil[3], Martin Bergert [4], Thibaut Sanchez [1], Arnaud Métais[1], Thomas Mangeat[5], Elisabeth Bellard[1,6], Claire Bigot[1], Catherine Astarie-Dequeker[1], Arnaud Labrousse [1], Jean-Philippe Girard[1,6], Isabelle Maridonneau-Parini[1], Christel Vérollet [1], Frédéric Lagarrigue[1], Alba Diz-Muñoz [4], Julien Heuvingh [3], Matthieu Piel[7], Olivia du Roure [3], Véronique Le Cabec [1✉], Sébastien Carréno [2✉] & Renaud Poincloux [1✉]

## Abstract

**The cellular cortex provides crucial mechanical support and plays critical roles during cell division and migration. The proteins of the ERM family, comprised of ezrin, radixin, and moesin, are central to these processes by linking the plasma membrane to the actin cytoskeleton. To investigate the contributions of the ERM proteins to leukocyte migration, we generated single and triple ERM knockout macrophages. Surprisingly, we found that even in the absence of ERM proteins, macrophages still form the different actin structures promoting cell migration, such as filopodia, lamellipodia, podosomes, and ruffles. Furthermore, we discovered that, unlike every other cell type previously investigated, the single or triple knockout of ERM proteins does not affect macrophage migration in diverse contexts. Finally, we demonstrated that the loss of ERMs in macrophages does not affect the mechanical properties of their cortex. These findings challenge the notion that ERMs are universally essential for cortex mechanics and cell migration and support the notion that the macrophage cortex may have diverged from that of other cells to allow for their uniquely adaptive cortical plasticity.**

**Keywords** Cell Migration; Cell Cortex; Cytoskeleton; ERM; Macrophages
**Subject Category** Cell Adhesion, Polarity & Cytoskeleton

## Introduction

The actin cytoskeleton plays a major role to support the cell shape changes necessary for diverse biological processes such as morphogenesis, mitosis, phagocytosis, and cell migration. The cellular cortex is organized in a thin actomyosin layer located just beneath the plasma membrane. The precise organization and regulation of this dynamic structure is critical for physiological processes, as disruptions of the actin cortex have been linked to different pathologies, such as cancer (Chugh and Paluch, 2018). Membrane-cortex attachment is essential for the regulation of mechanical properties and dynamics of the cortex, essential for processes such as mitosis (Carreno et al, 2008; Kunda et al, 2012, 2008; Leguay et al, 2022; Roch et al, 2010; Roubinet et al, 2011) or the formation of actin-driven and bleb-based protrusions during cell migration (Lembo et al, 2023; Welf et al, 2020). Ezrin, radixin and moesin compose the ERM protein family and are one of the main mechanical linkers between the plasma membrane and the actin cytoskeleton (Bretscher, 1983; Bretscher et al, 1997; Lankes and Furthmayr, 1991; Tsukita et al, 1989).

As such, ERM activation at mitotic entry was shown to promote rounding of metaphase cells by increasing stiffening of the cortex (Kunda et al, 2008; Larson et al, 2010; Leguay et al, 2022). By solidifying the actin-to-membrane link, ERMs also increase membrane tension in mesendodermal progenitors of zebrafish embryos, epithelial cells, and lymphocytes (Diz-Muñoz et al, 2010; Liu et al, 2012; Rouven Brückner et al, 2015; Tsujita et al, 2021). In addition, ERMs modulate the migratory speed of both normal and metastatic cells, either decreasing or increasing it (Li et al, 2017; Moodley et al, 2020; Tsujita et al, 2021). Having a key role in the link between the plasma membrane and the actomyosin cortex, ERMs control the organization and dynamics of thin actin structures such as brush border microvilli of epithelial cells (Bonilha et al, 2006; Crepaldi et al, 1997; Saotome et al, 2004) or filopodia of neural growth cones (Furutani et al, 2007; Gallo, 2008). On the contrary, ERM deactivation induces a low membrane-cortex attachment, promoting bleb formation in migrating zebrafish progenitors during gastrulation (Diz-Muñoz et al, 2016, 2010; Olguin-Olguin et al, 2021) or during division of drosophila cells (Kunda et al, 2008; Roubinet et al, 2011).

The individual roles of ezrin, radixin, and moesin have been studied without distinction, resulting in a limited understanding of

[1]Institut de Pharmacologie et de Biologie Structurale (IPBS), Université de Toulouse, CNRS, Université Toulouse III - Paul Sabatier (UT3), Toulouse, France. [2]Institut de Recherche en Immunologie et en Cancérologie (IRIC), Université de Montréal, Montréal, Canada. [3]PMMH, ESPCI Paris, PSL University, CNRS, Université Paris Cité, Sorbonne Université, Paris, France. [4]Cell Biology and Biophysics Unit, European Molecular Biology Laboratory, Heidelberg, Germany. [5]LITC Core Facility, Centre de Biologie Intégrative, Université de Toulouse, CNRS, UPS, 31062 Toulouse, France. [6]Equipe Labellisée Ligue Contre le Cancer, Paris, France. [7]Institut Curie et Institut Pierre Gilles de Gennes, PSL University, CNRS, Paris, France. ✉E-mail: veronique.le-cabec@ipbs.fr; sebastien.carreno@umontreal.ca; renaud.poincloux@ipbs.fr

their unique functions within the same cell. To bridge this knowledge gap, we aimed to investigate the specific contributions of each ERM to macrophage migration. Macrophages are innate immune cells that represent the archetype of the plastic cell capable of infiltrating every tissue of the organism to maintain immune surveillance and tissue homeostasis. Unlike other leukocytes, which rely solely on the protease-independent ameboid mode of migration (Cougoule et al, 2010; Lämmermann et al, 2008), macrophages adapt their motility strategy to their external environment. In addition to the ameboid mode, macrophages are capable of infiltrating dense matrices by using a protease-dependent mesenchymal migration that relies on adhesive podosome actin structures (Gui et al, 2018; Van Goethem et al, 2011, 2010). Surprisingly, we found that ERMs are dispensable for macrophages to migrate in diverse contexts, including in vitro 2D migration and 3D invasion of extracellular matrix, ex vivo tissue infiltration through healthy dermis and tumor tissue, and for the in vivo adhesion of macrophage precursors to an activated endothelium. We further show that neither the thickness nor the mechanics of the actin cortex of macrophages are affected by ERM knockout.

## Results

### The deletion of ezrin, radixin, or moesin does not affect macrophage 3D migration

In order to investigate the function of ERM proteins in macrophages, we first determined which ERM are expressed in these cells. We confirmed that moesin (Msn) and ezrin (Ezr), but not radixin (Rdx) are expressed in monocytes (Shcherbina et al, 1999). We also showed that all three ERMs are expressed in human blood monocytes-derived macrophages (HMDM) (Fig. 1A,B). ERMs were also found in macrophages in their phosphorylated, active form (P-ERM), that links the plasma membrane to the actin cytoskeleton (Bretscher et al, 1997; Matsui et al, 1998; Pelaseyed et al, 2017).

We then investigated the localization of ERM proteins in macrophages. For this purpose, we transfected ezrin-, radixin-, and moesin-GFP into human monocyte-derived macrophages (HMDM). All three ERMs localized predominantly at the plasma membrane and decorated membrane ruffles (Fig. EV1; Movies EV1–6). Of note, ezrin, but neither radixin nor moesin, was also slightly enriched around podosome cores (Fig. EV1A; Movies EV1 and 4). Live observations also revealed that nascent, peripheral, ruffles were enriched in actin filaments, whereas Ezrin, Radixin, and Moesin accumulated indifferently on nascent and older, more central, ruffles (Fig. EV1D; Movies EV4–6).

Moesin being described as the most abundant ERM expressed in leukocytes (Shcherbina et al, 1999), we then knocked down the expression of moesin in HMDM using small interfering-RNAs (siRNA). siRNA treatments resulted in an 80% ± 5% moesin depletion (mean ± SD), (Fig. EV2A). Surprisingly, we found that moesin-depleted human macrophages and wild-type HMDMs exhibited similar levels of infiltration into 3D fibrillar collagen I, an extracellular matrix that promotes the ameboid mode of motility in macrophages (Van Goethem et al, 2010) (Fig. EV2B). We also observed that in Matrigel, which promotes protease-dependent

mesenchymal migration of macrophages (Van Goethem et al, 2010), moesin is dispensable for these cells to infiltrate this extracellular matrix (Fig EV2C). Since ezrin and radixin could also potentially replace moesin loss, we decided to knock down all ERMs using siRNAs. siRNA treatments resulted in a depletion of 59% ± 28%, 81% ± 10% and 61% ± 20% for ezrin, radixin, and moesin, respectively (mean ± SD) (Fig. 1C). Surprisingly, triple ERM depletion did not affect macrophage migration through either collagen I or Matrigel (Fig. 1D,E).

Yet, as a small residual amount of ERMs after siRNA may be sufficient to ensure their function, we then decided to knockout each ERMs using CRISPR–Cas9 gene edition. We took advantage of an HoxB8-immortalized murine line of macrophage progenitors, which allows an easy and fast generation of progenitors genetically depleted for any gene of interest (Accarias et al, 2020). We first demonstrated that all three ERMs are expressed in mouse HoxB8 macrophages (Fig. 1F,G), moesin being the most expressed and phosphorylated ERM (Fig. 1F). We first knocked out ezrin, radixin, or moesin individually in HoxB8 progenitors by CRISPR–Cas9. As a control, we used sgRNA targeting the non-expressed luciferase protein, hereafter referred as WT. We achieved a depletion of 89%, 85%, and 97% for ezrin, radixin, or moesin, respectively, in multiclonal populations of macrophages individually knocked out for ERMs (Fig. 1H). Notably, moesin was the main expressed ERM in macrophages, as evidenced by the lack of detectable western blot signal for ERMs using the pan-ERM antibody in moesin knockout HoxB8 macrophages. Moreover, among the three ERMs, only moesin knockout resulted in a significant decrease in P-ERM, suggesting that moesin is also the main activated ERM in HoxB8 macrophages.

Our findings were again unexpected as we discovered that the individual knockout of any ERM did not affect the 3D migration of macrophages in either collagen I nor in Matrigel (Fig. 1I,J). It is possible that a compensation mechanism took place during the selection and amplification of ERM-KO cells. To minimize this possibility and reduce the interval between gene deletion and migration assays, we generated five new and independent moesin KOs immediately prior to each migration experiment, without any cell amplification. Subsequent analyses revealed that these KOs had no effect on macrophage migration in 3D (Fig. EV2D–F), which suggests that if a compensatory mechanism takes over, it is instantaneous and does not require gene expression regulation.

Since the three ERM proteins exhibit high structural similarity and function, it is possible that the other ERMs in each single knockout compensate for the loss of the knocked-out ERM. To test this hypothesis, we decided to simultaneously knock out all three ERM proteins in HoxB8-macrophage progenitors.

### The triple KO of ezrin, radixin, and moesin does not significantly affect the formation and dynamics of actin structures in macrophage

We generated a triple knockout of the ezrin, radixin and moesin proteins in HoxB8 myeloid progenitors. We isolated three different clonal populations where all three ERM proteins were fully depleted, as validated by immunoblot analysis (Fig. 2A), and genomic DNA sequencing (Appendix Fig. S1), and refer to them as ERM-tKO (clones #1, #2 and #3). To control for potential off-target effects of CRISPR–Cas9 editing, we selected three independent

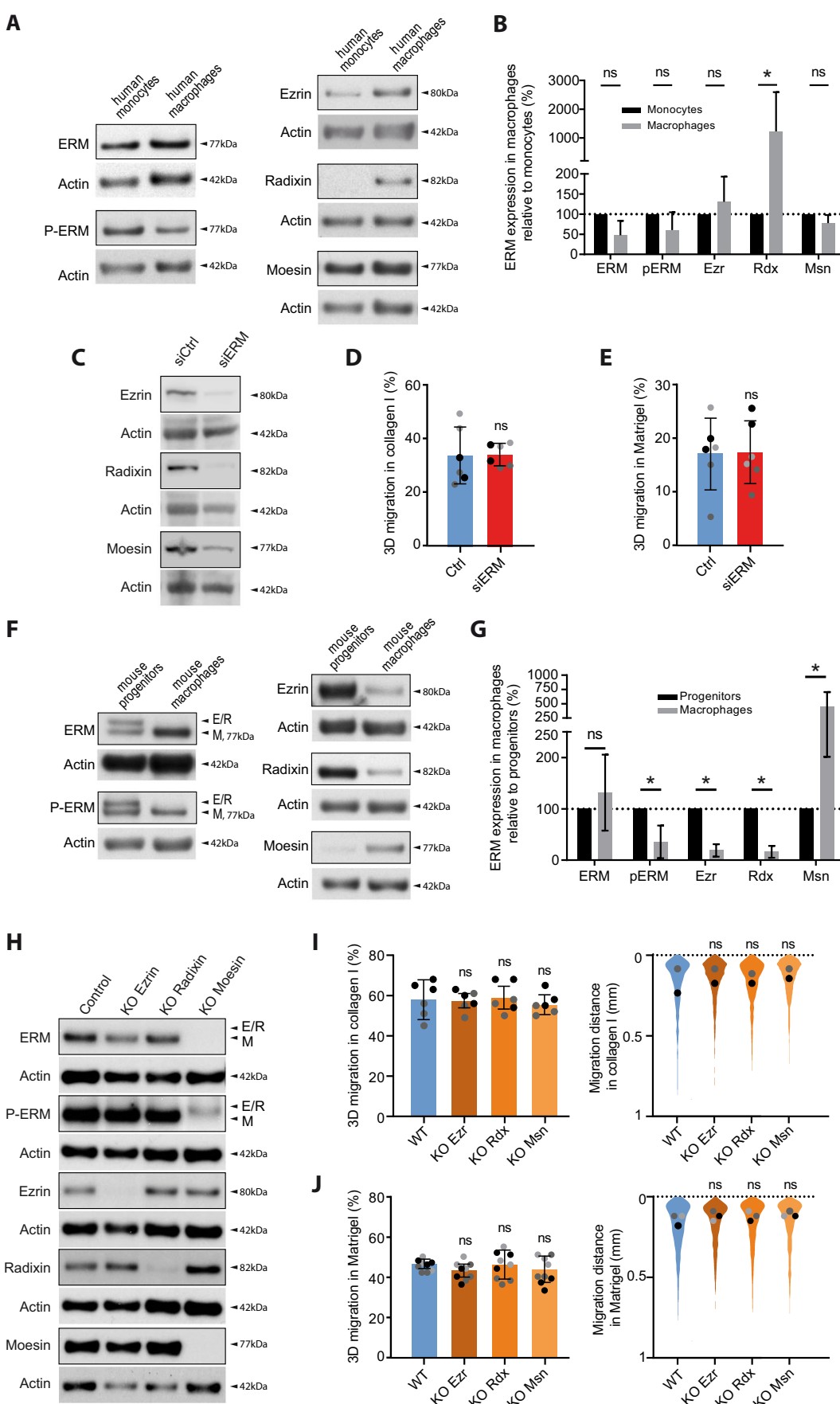

◄ **Figure 1. Ezrin, radixin, or moesin depletion does not affect macrophage 3D migration.**

(A, B) Endogenous expression of ERM proteins in human myeloid cells. (A) ERM, p-ERM, ezrin, radixin, and moesin expression levels of human blood-derived monocytes, and human monocyte-derived macrophages (HMDM), respectively, differentiated from the same donor. Actin levels were used to normalize the immunoblots. (B) Quantification of ERM expression in macrophages relative to monocytes was done at least on three independent donors. Mean and SD are plotted. Statistics: P values were obtained with a paired Student's t test (ns: not significant, *P < 0.05). P value (Rdx)= 0.0357. (C–E) ERM depletion in human macrophages by siRNA. (C) Ezrin, radixin, and moesin expression levels of HMDM treated with siCtrl or siRNAs against the three ERMs is representative of three independent donors. (D, E) Percentages of migration of siRNA-treated HMDM inside collagen I (D) and Matrigel (E) are represented as follows: the technical replicates (dot) of three independent experiments (highlighted by different gray colors) are represented. The mean (bar) and SD from the independent experiments are shown. Statistical analysis was done on the mean per experiment using a paired two-tailed t test. (F–H) Endogenous expression and depletion of ERM in murine macrophages. (F) ERM, p-ERM, ezrin, radixin, and moesin expression levels of murine WT HoxB8 progenitors, and WT HoxB8 macrophages. (G) Quantification of expression relative to HoxB8 progenitors from at least three independent experiments. Mean and SD are plotted. Statistics: P values were obtained with a paired Student's t test. P value (p-ERM)=0.0325; (Ezr)=0.0004; (Rdx)=0.0003; (Msn) =0.0027. (H) Immunoblots showing the expression levels of ERM, p-ERM, ezrin, radixin and moesin of HoxB8 macrophages WT or respectively knockout for ezrin (KO Ezrin), radixin (KO Radixin) or moesin (KO Moesin). (I, J) 3D migration of simple KO of ezrin-, radixin-, or moesin in macrophages: percentages of migration and migration distances of WT, KO Ezrin, KO Radixin, or KO Moesin HoxB8 macrophages inside collagen I (I) and Matrigel (J) are plotted as follows: the technical replicates (dot) of three independent experiments (highlighted by different gray colors) are represented. The mean (bar) and SD from the three independent experiments are shown. Statistical analysis was done on the mean per experiment using a RM one-way ANOVA. The distribution of the migration distance of each cell from three independent experiments is shown. Dots represent the median of each independent experiment and were used for statistical analysis using RM one-way ANOVA. Source data are available online for this figure.

clones that had different genomic mutations in their ezrin, radixin, and moesin genes.

As ERMs were found to control the formation and dynamics of diverse actin protrusions (Brown et al, 2003; Furutani et al, 2007; Saotome et al, 2004), we first quantified the formation of these actin structures in ERM-tKO macrophages and progenitors (Fig. 2B–G). We observed that both WT and ERM-TKO progenitors formed a similar number of microvilli-, filopodia- and ruffle-like short protrusions on glass coverslips, as revealed by scanning electron microscopy (SEM) (Fig. 2B).

We then found that the ERM-tKO macrophages spread efficiently on bare glass surface in a similar manner than WT macrophages, as demonstrated by their comparable cell area (WT: $916 \pm 27\,\mu m^2$; TKO: $884 \pm 32\,\mu m^2$, mean ± SEM) and circularity (WT: $0.18 \pm 0.007$; TKO: $0.19 \pm 0.008$) compared to the WT (Fig. 2D). We also measured that they form the same number of large lamellipodia (Fig. 2C), and thin filopodia protrusions (WT: $10.7 \pm 1.2$ filopodia per cell; TKO: $9.8 \pm 1.6$), although the latter are slightly longer in the ERM-tKO macrophages ($4.9 \pm 0.25\,\mu m$, mean ± SEM, compared to $4.0 \pm 0.23\,\mu m$ in WT macrophages) (Fig. 2E). The number of podosomes was also slightly reduced in the ERM-tKO ($50 \pm 2$ podosomes/cell; mean ± SEM) compared to WT macrophages ($69 \pm 3$) (Fig. 2F), but their dynamics remained unaffected (Fig. 2G; Movie EV7).

Overall, the absence of ERM proteins in macrophages does not impair the ability of these cells to form the various actin structures involved in migration. Although we observed a slight reduction in the density of podosomes and a slight increase in the length of filopodia, these alterations do not substantially affect the overall formation of actin cortical structures in ERM-deficient macrophages. This contrasts with the microvilli collapse observed in ezrin/moesin double KO in neutrophils and lymphocytes from Moesin$^{Y/-}$ mice (Brown et al, 2003; Hirata et al, 2012; Panicker et al, 2020), and suggests that these small protrusions might not have the same molecular composition in lymphocytes and neutrophils compared to monocyte progenitors.

To investigate whether these modest alterations could affect macrophage phagocytosis and migration, we first investigated the ability of ERM-deficient macrophages to perform phagocytosis. Indeed, it has been suggested that ERMs play important roles

during phagocytosis (Di Pietro et al, 2017; Erwig et al, 2006; Gomez and Descoteaux, 2018). We therefore quantified the dynamics of phagocytosis of ovalbumin-coated or IgG-opsonized polystyrene beads, which did not reveal any difference between WT and ERM-depleted macrophages (Fig. EV3).

We then tested wether ERM-tKO affect the macrophage capacity to migrate randomly on 2D glass, or in response to a gradient of complement 5a (C5a) as a chemoattractant. We found that there were no significant differences in the average velocity (means of WT, 2.6 μm/min; tKO#1, 2.45 μm/min; tKO#2, 2.62 μm/min; tKO#3, 2.36 μm/min) and confinement ratio (WT: $0.20 \pm 0.02$; tKO#1, $0.14 \pm 0.02$; tKO#2, $0.18 \pm 0.02$; tKO#3, $0.19 \pm 0.01$; mean ± SEM) of the 3 ERM-tKO clones and the WT macrophages (Fig. 2H; Movie EV8). ERM-tKO macrophages were also perfectly able to perform chemotaxis toward C5a without any defects in their velocity, directionality or their chemotaxis index (Fig. EV4; Movie EV9).

## The 3D infiltration of macrophages in vitro remains unaffected by the loss of ezrin, radixin, and moesin

As macrophages migrate in vivo mostly in a tridimensional context, we next aimed to test whether ERMs could regulate 3D migration. We previously reported that when invading collagen I, macrophages rely solely on the protease-independent ameboid mode of migration while they use protease-dependent mesenchymal migration when they invade Matrigel. We found that ERMs were dispensable for both modes of invasion. All three ERM-tKO clones infiltrated both collagen I (Fig. 3A,C) and Matrigel (Fig. 3B,D) similarly to WT macrophages. Live imaging of WT and ERM-tKO macrophages in 3D collagen I and Matrigel confirmed that ERM depletion did not affect invasion of macrophages (Movies EV10–13). In addition, using live super-resolution imaging of WT and ERM-tKO macrophages stably expressing lifeact-GFP, we found that in 3D collagen, the ruffles formed by tKO macrophages retract at the same rate than WT macrophages, showing that ERMs are not necessary for the dynamics of these actin structures (Fig. 3E–G; Movies EV14 and 15).

When differentiated into pre-osteoclasts, no significant effect on migration through the two matrices was observed either. Importantly,

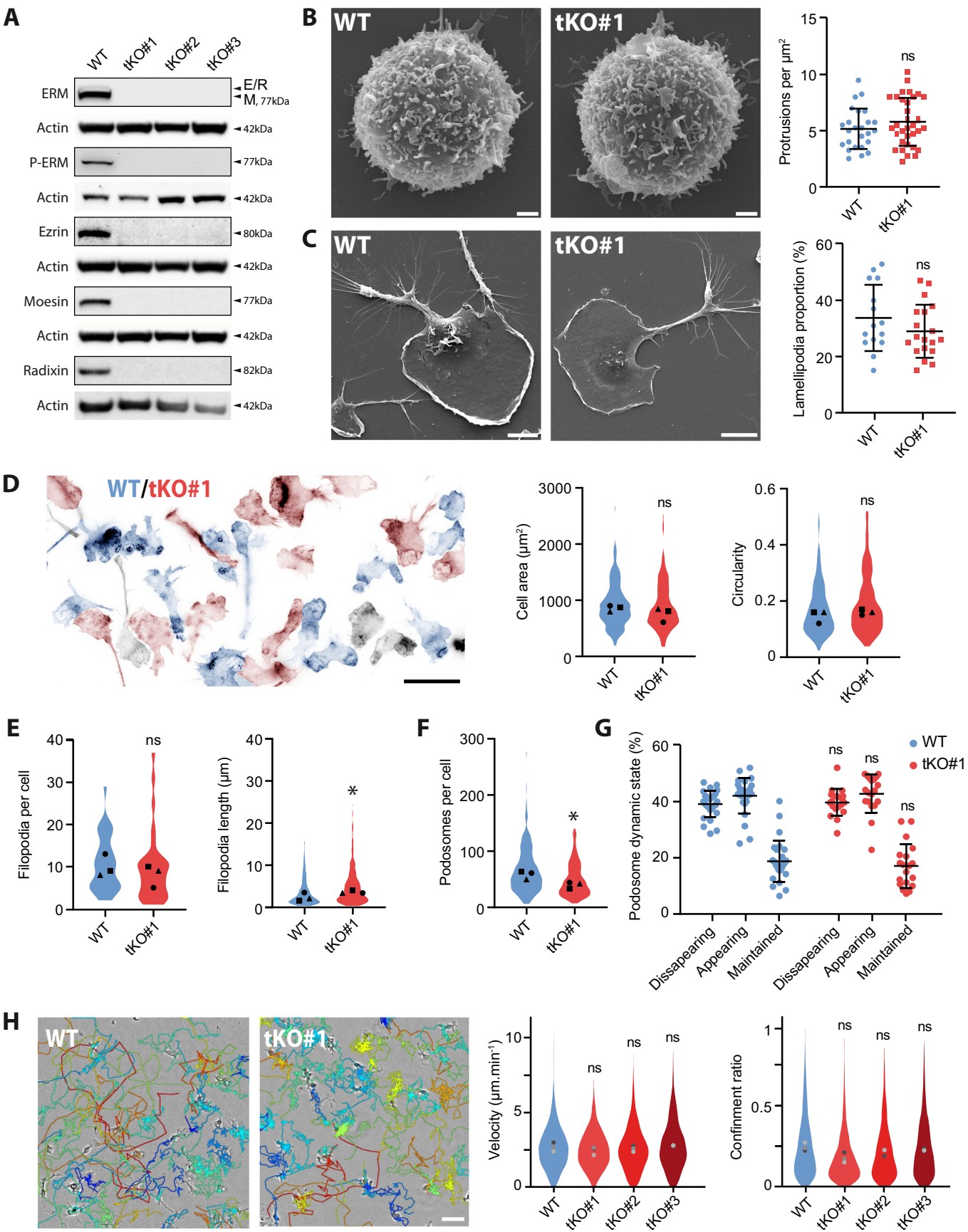

**Figure 2. ERM-tKO does not have a marked influence on the formation and dynamics of macrophage actin structures.**

(A) The expression levels of ezrin, radixin, moesin, ERM, and p-ERM from WT Hoxb8 macrophages and three independent clones (tKO#1, tKO#2, tKO#3) triple ERM knockout (ERM-tKO) were analyzed by western blot. ERM and moesin are from the same immunoblots, that were stripped and rehybridized, and therefore have the same actin blot. (B–G) Morphological analysis of ERM-tKO macrophages. (B) The morphology of WT and ERM-tKO#1 progenitors was analyzed by scanning electron microscopy (SEM) and the number of microvilli-like protrusions was quantified per μm$^2$ in 25 WT and 34 ERM-tKO#1 progenitors from three independent experiments (right panel). Scale bar: 1 μm. Statistics (B, C): means and SD are plotted and $P$ values were obtained with a Mann–Whitney $U$ test on all cells (ns: not significant, *$P < 0.05$). (C) Morphology of WT and ERM-tKO#1 macrophages was analyzed by SEM and lamellipodia were evaluated as a proportion of the cell perimeter in 15 WT and 19 ERM-tKO#1 cells from two independent experiments (right panel). Means and SD are plotted. Scale bar: 10 μm. (D) Actin staining of WT (blue) and ERM-tKO#1 (red) macrophages pseudo-colored according to their identification using cell trackers. Black cells, whose cell tracer staining was too weak to be identified, were removed from the analysis (left panel). Quantification of cell area (in μm$^2$) and circularity of 210 WT and 162 ERM-tKO#1 macrophages (right panels). Statistics (D–F): the medians of three independent experiments are represented. A Mann–Whitney $U$ test was used on all cells for statistical analysis (ns: not significant, *$P < 0.05$). Scale bar: 20 μm. (E) Quantification of the number of filopodia per cell in 30 WT and 30 ERM-tKO#1 macrophages (left panel), and the associated filopodia length from 320 WT and 294 ERM-tKO#1 filopodia of macrophages (right panel) from two independent experiments. Filopodia length $P$ value = 0.0002. (F) Quantification of podosomes number of 210 WT and 162 ERM-tKO#1 HoxB8-macrophage cells. $P$ value < 0.0001. (G) Quantification of podosome stability from lifeact-GFP-expressing WT and ERM-tKO#1 macrophages plated in 2D bare glass and imaged with RIM. See also Movie EV7. The dynamic states of podosomes were categorized into disappearing, appearing and maintained and expressed as percentages. Statistics were evaluated on 29 WT and 19 ERM-tKO#1 cells from two independent experiments using with RM one-way ANOVA. Means and SD are shown. See Appendix Fig. S2 for detailed explanations. (H) 2D Migration of WT, ERM-tKO#1, tKO#2, and tKO#3 macrophages. Snapshot pictures showing WT and ERM-TKO#1 macrophages migrating randomly in bare glass (2D) with migratory tracks representing cell trajectories during 10 h. Tracks are color-coded according to their mean speed. Scale bar: 50 μm. See also Movie EV8. Quantification of the median velocity and the confinement ratio (0: confined movement; 1: directionally persistent movement) of each migratory track from WT, ERM-tKO#1, tKO#2, and tKO#3 HoxB8 macrophages. Violin plots represent the distribution of the analyzed parameter for all the filtered migratory tracks, and the medians of three independent experiments are represented and used for statistical analysis with RM one-way ANOVA. Source data are available online for this figure.

when the same ERM-tKO progenitors were differentiated with GM-CSF into dendritic-like cells, characterized by CD11b+, CD11c high, F4/80−, and Ly6C− markers, a 1.8-fold increase in their capacity to migrate through Matrigel was observed, whereas their capacity to migrate through Collagen I remained unaffected (Fig. EV5). These results suggest that the lack of a role for ERMs is specific to the migration of macrophages and pre-osteoclasts, and not shared among all myeloid cells.

## ERM-tKO cells without ezrin, radixin, and moesin exhibit no impairment in their ability to adhere to vascular endothelium in vivo and infiltrate the ear derma or fibrosarcoma

To further investigate the migratory properties of ERM-deficient cells in vivo, we first assessed their ability to adhere to activated vascular endothelium into mice bearing a fibrosarcoma (Gui et al, 2018). Indeed, ERM-deficient neutrophils have been described to be impaired for their capacity to adhere to an endothelium (Matsumoto and Hirata, 2016; Panicker et al, 2020). For that purpose, one-day differentiated wild-type or ERM-deficient cells were fluorescently labeled with two different cell trackers, mixed in a 1:1 ratio, and co-injected intra-arterially into recipient mice in order to analyze their behavior in tumor blood vessels by intravital microscopy. Using real-time intravital imaging, we assessed the proportion of cells rolling along the tumor vessels and sticking to it. We revealed that the rolling and sticking fractions of ERM-tKO were similar to those of WT cells (Fig. 4A).

We then demonstrated that ERMs are dispensable for the interstitial infiltration of macrophages into various tissues as different as the ear dermis and tumor tissue ex vivo. To model interstitial tissue migration, we developed an ex vivo migration assay in the mouse ear dermis, in which we allowed differentially labeled WT and ERM-tKO macrophages to infiltrate for three days. To overcome the high heterogeneity of the thin ear tissue, we placed both cell types in the same tissue. Additionally, we switched the cell trackers used to stain the cells between experiments to

verify the absence of effects due to staining. Tissue sections followed by immunohistochemistry demonstrated that both tKO and WT macrophages infiltrate the ear derma without any significant difference (Fig. 4B). Finally, we evaluated whether ERM-deficient macrophages could infiltrate tumors. As previously described (Accarias et al, 2020; Gui et al, 2018), we generated fibrosarcoma by subcutaneous injection of LPB tumor cells in mice, and the tumors were resected and sliced into thick explants of 500 μm height. A 1:1 mix of cell tracker-labeled WT and tKO-ERM macrophages was seeded on top of the explants, and the infiltration of the macrophages into the tumor tissue was monitored after 3 days. We found that tKO-ERM macrophages infiltrated the tumor tissue in the same proportion and at the same distance as the WT macrophages (Fig. 4C).

## ERMs do not control the mechanical properties of the macrophage cortex and are dispensable for membrane-to-cortex attachment (MCA)

Our results demonstrate that ERM proteins, which are recognized for playing crucial roles in the actin cortex (Diz-Muñoz et al, 2010; Welf et al, 2020), are not required for macrophage migration. We therefore questioned whether ERMs are actually required for the organization of the actin cortex in macrophages by assessing the size and mechanics of these cortexes. Specifically, we used phagocytosis to internalize magnetic beads within WT and tKO-ERM macrophages, and then employed a magnetic field to align the internalized beads with those present in the extracellular environment. By pinching the cortex between the internalized and external beads, we were able to directly measure its thickness and rigidity (Fig. 5A–C) (Laplaud et al, 2021). Demonstrating that ERMs are dispensable for the organization of the cortex of macrophages, we did not observe any differences between ERM-tKO and WT macrophages. The thickness of the ERM-tKO cortex (252 ± 29 μm; mean ± SEM) was similar those of WT macrophages (284 ± 29 μm) (Fig. 5D), and when submitted to gradually increasing stresses, we did not measure any differences between the response of the

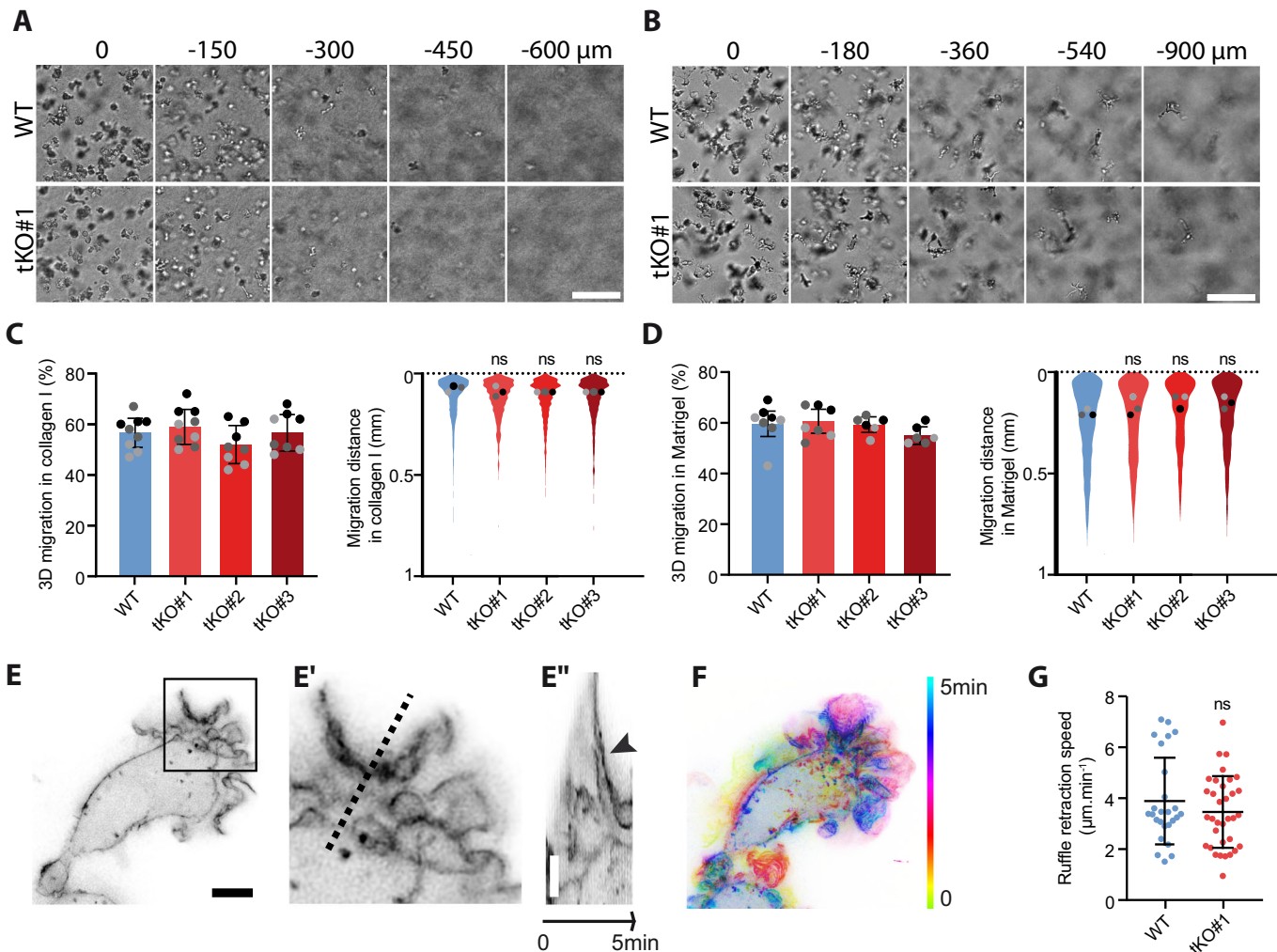

**Figure 3. ERMs are dispensable for macrophage infiltration through 3D matrices.**

(A, B) Morphology of WT and ERM-tKO#1 Hoxb8 macrophages inside 3D collagen I (A) or 3D Matrigel (B) is shown. Scale bars: 100 µm. See also z-stacks Movies EV10 and 11, respectively, in 3D Collagen I and 3D Matrigel, as well as time-lapse Movies EV12 and 13. (C, D) Percentages of migration and migration distances of WT, ERM-tKO#1, tKO#2, and tKO#3 HoxB8 macrophages in collagen I (C) and Matrigel (D) are represented as follows: the technical replicates (dot) of three independent experiments (highlighted by different colors) are represented. The mean (bar) and SD from the three independent experiments are shown and used for statistical analysis using RM one-way ANOVA. The distribution of the migration distance of each cell from three independent experiments is shown. Dots represent the median of each independent experiment and were used for statistical analysis using RM one-way ANOVA. (E, F) Acquisition of a Lifeact-GFP WT macrophage in 3D collagen I (E), and time-lapse color-coded from 0 to 5 min for the dynamics (F). See also Movie EV14 to compare WT and TKO cells in collagen I. Kymograph (E″), following the pointed line in enlarged view (E′), shows the dynamics of a ruffle (arrowhead). Scale bars: 5 µm. (G) Quantification of ruffle retraction in the 3D collagen I matrix from 26 measures of WT and 34 of ERM-tKO#1 macrophages expressing lifeact-GFP and imaged with RIM. See also Movie EV14. In total, 26 ruffles from 19 WT cells and 34 TKO#1 cells from two independent experiments. Unpaired t test P = 0.288. Means and SD are shown. Source data are available online for this figure.

WT (4544 ± 662 Pa; mean ± SEM) and ERM-tKO macrophages (6574 ± 1259 Pa) subjected to stresses of gradually increasing intensity (Fig. 5E).

Given the absence of structural and mechanical differences in the macrophage cortex upon ERM depletion, we also asked if the physical link between the actomyosin cortex and the plasma membrane (membrane-to-cortex attachment, MCA) is affected. We thus performed dynamic tether pulling using atomic force spectroscopy to quantify MCA in CTL and ERM-TKO HoxB8 macrophage (Fig. 5F). In line with the magnetic pinching experiments, we found MCA not to be altered upon ERM depletion (Fig. 5G,H). Thus, ERMs seem to be dispensable in macrophages

for their canonical function, which is to mechanically link the actomyosin cortex and the plasma membrane.

Finally, to further investigate the functional role of ERM proteins for the dynamic mechanical properties of the macrophage cortex and its attachment to the plasma membrane, we conducted experiments on membrane blebbing. Membrane blebs play an important role in cell migration, particularly in 3D environments, and are promoted by hydrostatic pressure from the cytoplasm that causes cortical actin rupture and plasma membrane protrusions. To retract, blebs require a mechanical link between actomyosin contractions and the plasma membrane of their rim. Interestingly, in all cells studied so far, this mechanical link is strictly dependent

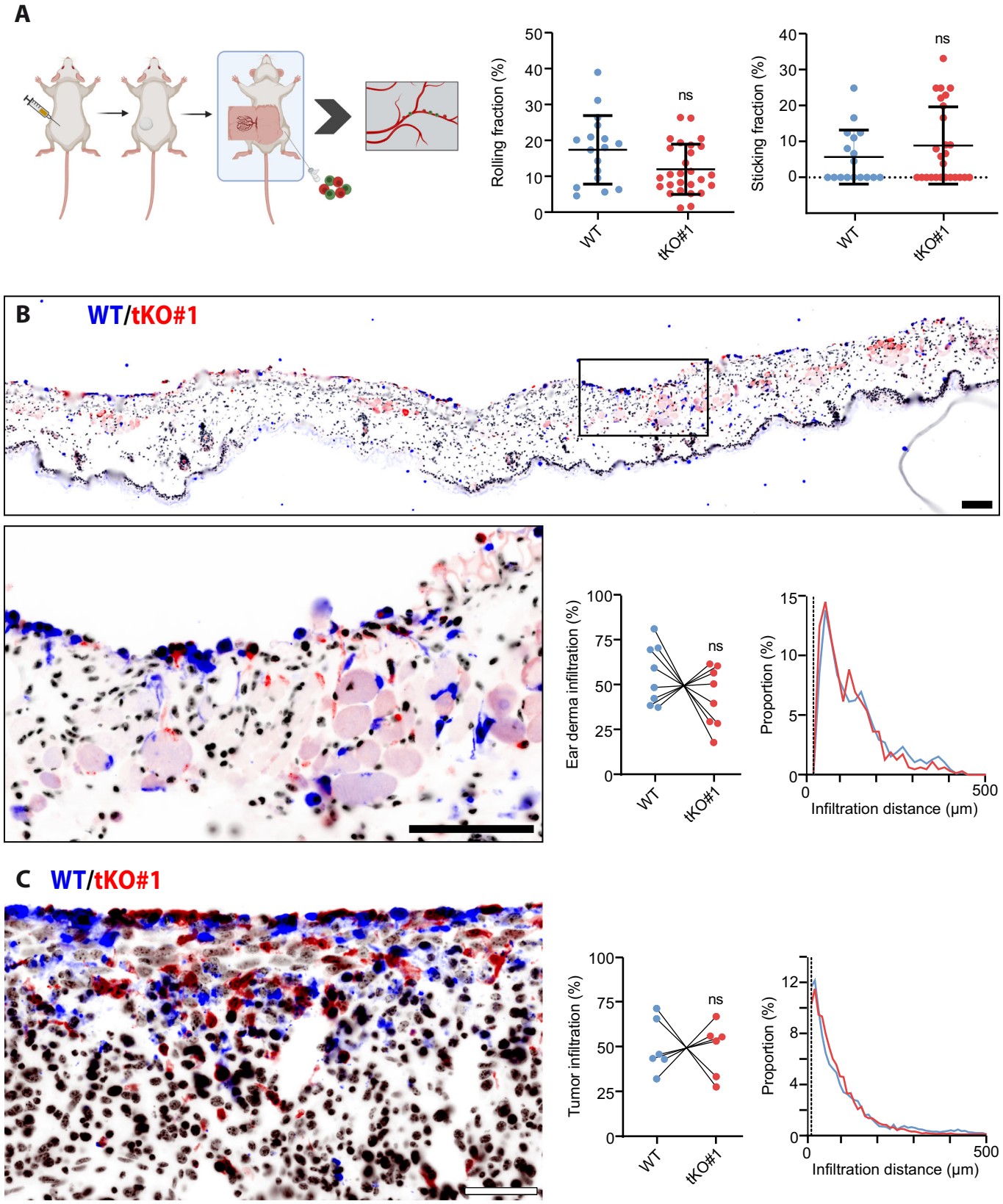

◄ **Figure 4.   ERM-tKO cells have no defect in adhesion to vascular endothelium in vivo and infiltrate tissue explants ex vivo.**

(A) In vivo adhesion to vascular endothelium. Fibrosarcoma cells were injected into the flank of a mice. After a week, tumor was exposed for intravital microscopy, and the femoral artery of recipient mice was catheterized for injection of exogenous cells. Differentially labeled WT and TKO-ERM macrophage precursors were injected in the blood and their behavior in tumor blood vessels was assessed by real-time imaging. Rolling fractions were quantified as the percentage of rolling cells in the total flux of cells in each blood vessel, and sticking fractions were quantified as the percentage of rolling cells that firmly adhered for a minimum of 30 s. In total, 19 and 28 blood vessels were used as replicates for WT and TKO#1 precursor cells, respectively, on four different mice. *P* values were assessed with a Mann–Whitney test. Means and SD are represented. (B) Ex vivo infiltration of ear derma. Differentially labeled WT and tKO#1 macrophages were seeded on top of a murine ear derma tissue over 3 days. Slices were then fixed and serial sectioning was performed along the *z* axis. Immunohistofluorescence of an ear section showing WT (blue) and tKO#1 (red) macrophage infiltration and dapi staining of all nuclei (black). Quantifications of the percentage and the distance of ear derma infiltration of WT or tKO#1 macrophage is represented as the mean of the respective infiltration percentage per ear halve section. Analysis was performed on seven independent ear halve explants from four mice. Scale bars: 50 μm. (C) Ex vivo infiltration of tumor explants. Differentially labeled WT and tKO#1 HoxB8-macrophage cells were seeded on top of sliced fibrosarcoma explants over 3 days. Slices were then fixed and serial sectioning was performed along the *z* axis. Immunohistofluorescence of a tumor section showing WT (blue) and tKO#1 (red) macrophage infiltration and dapi staining of all nuclei (black). Quantifications of the percentage and the distance of tumor infiltration of WT or tKO#1 macrophages are represented. Means of six ex vivo independent tumor explants from three tumors are shown. Representative pictures are shown. Scale bars: 50 μm. Of note, cell trackers used to stain the cells were switched between all experiments to verify the absence of effects due to staining. Source data are available online for this figure.

on ERM activity (Charras et al, 2006; Diz-Muñoz et al, 2010; Roubinet et al, 2011). Using time-lapse microscopy to monitor bleb formation, we found that ERMs are unexpectedly dispensable for bleb dynamics in macrophages. We found no significant difference in maximum bleb size and bleb retraction rates between WT and ERM-tKO macrophages (Fig. 5I–K; Movie EV16).

## Discussion

ERMs are the main linker between the actin cytoskeleton and the plasma membrane, and as such, they play a crucial role in various cellular functions involving actin cortex remodeling (Fehon et al, 2010). For instance, these proteins are pivotal in controlling the motility of diverse cell types, including cancer cells (Barik et al, 2022).

Our study uncovered a surprising finding: ezrin, radixin and moesin appear to be dispensable for key aspects of macrophage behavior, involving actin cortex remodeling, such as the formation of lamellipodia and filopodia, the dynamics of membrane ruffles and podosomes, phagocytosis, migration in vitro (in 2D or 3D matrices) and ex vivo (into dermis or tumor tissues) as well as for the in vivo adhesion of macrophage precursors to activated vascular endothelium. Furthermore, we demonstrated that ERMs are dispensable for several mechanical properties of the macrophage cortex, including cortical thickness and stiffness, membrane-to-cortex attachment and the control of bleb dynamics.

Although our findings indicate that ERMs are not necessary for the standard functions of macrophages, we cannot completely discount that ERMs are necessary under specific conditions. Supporting this hypothesis, recent studies reported that a triple ERM knockout in RAW macrophages did not result in global morphological changes. Phagocytosis remained unchanged between control and triple ERM knockout RAW macrophages. However, an increase in CD44 mobility and an inhibition of podosome disappearance concomitant with IgG phagocytosis were reported in Raw ERM-KO macrophages (Ferling et al, 2024; Le et al, 2024). This suggests that ERMs, although not important for macrophage migration and phagocytosis may play a role in coordinating various macrophage functions.

Despite observing no differences in the behavior of ERM-tKO compared to control macrophages across a wide array of assays, the absence of phenotypes upon ERM knockouts could be due to compensation by other proteins during the establishment of

ERM-tKO HoxB8 cells. However, reducing the time between depletion of ERMs, either by siRNA or by CRISPR–Cas9, still did not yield any observable effects. This suggests that the absence of ERMs is unlikely to be compensated for by a gene regulation mechanism.

In any case, this study challenges the widely held belief that ERMs are universal regulators of cortical organization and cell migration (Akisawa et al, 1999; Arpin et al, 2011; Crepaldi et al, 1997; Louvet-Vallée, 2000; Valderrama et al, 2012). Instead, these findings suggest that the macrophage actin cortex is highly adaptable and plastic, enabling these immune cells to function independently of ERM proteins. Macrophages demonstrate unique plasticity among leukocytes, especially in their capacity to adapt their migration mode to the microenvironment (Cougoule et al, 2012; Van Goethem et al, 2010). Our results suggest that, in contrast to macrophage migration, DC migration is affected by ERM inhibition (Fig EV5B,C), highlighting the specificity of macrophages even among their myeloid counterparts.

This adaptability may include the ability to employ alternative actin-membrane linkers, such as myosin 1a (Mazerik et al, 2014; Tyska et al, 2005), myosin 1b (Diz-Muñoz et al, 2010), myosin 1e (Gupta et al, 2013; Ouderkirk and Krendel, 2014) and myosin 1f (Navinés-Ferrer and Martín, 2020). In accordance with this hypothesis, while we demonstrated that ERMs are not involved in macrophage phagocytosis, recent studies have identified the involvement of myosins 1e and 1f in this process (Barger et al, 2019; Vorselen et al, 2021).

Our work paves the way for future investigations into how macrophages adjust their cellular architecture to meet various environmental demands, potentially leading to new insights into immune system behavior.

## Methods

### Differentiation and culture of human primary monocyte-derived macrophages

Monocytes from healthy subjects (HS) were provided by Etablissement Français du Sang, Toulouse, France, under contract 21/PLER/TOU/IPBS01/20130042. According to articles L12434 and R124361 of the French Public Health Code, the contract was approved by the French Ministry of Science and Technology (agreement number AC 2009921). Written informed consents were obtained from the

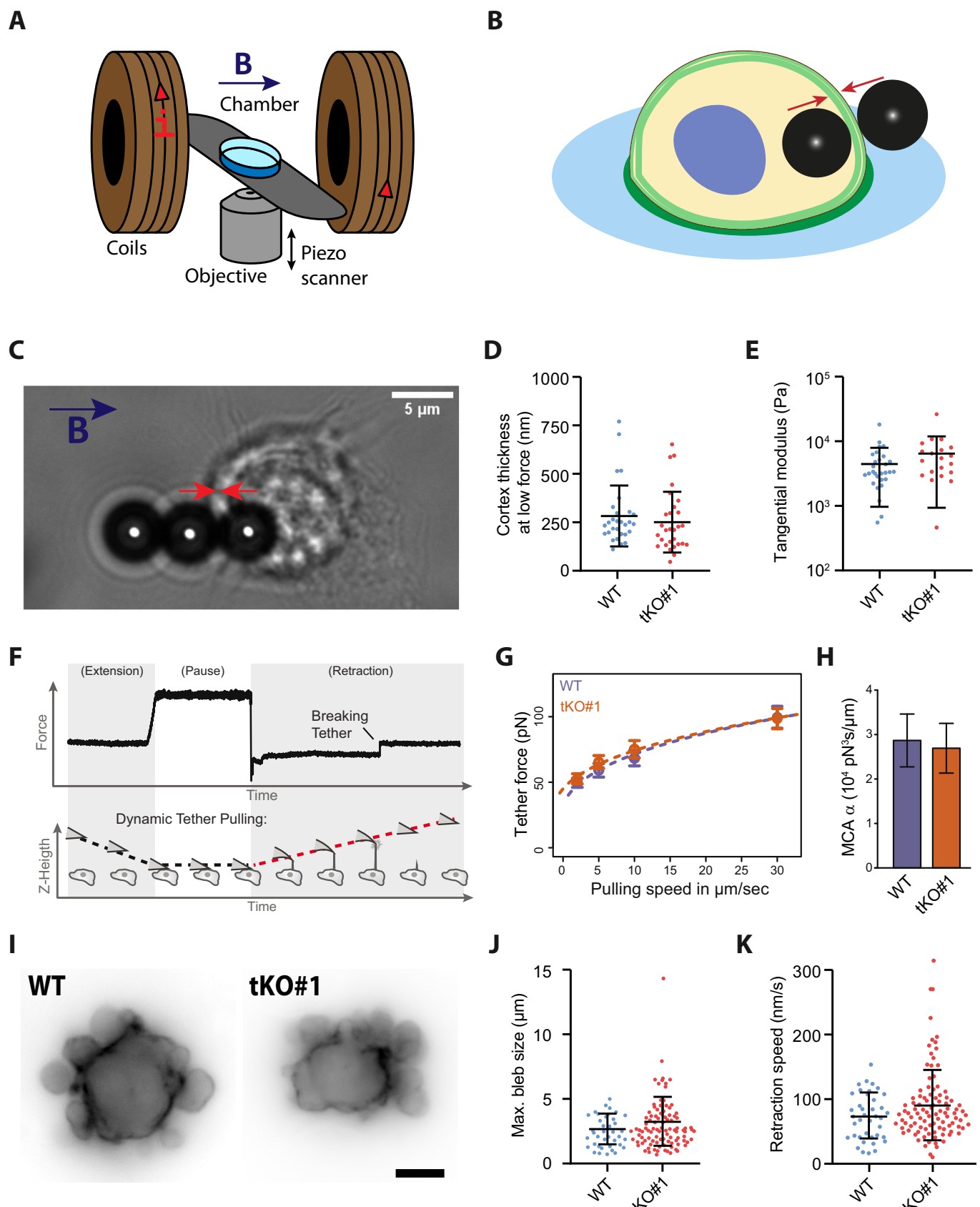

**Figure 5. ERM depletion does not affect macrophage cortex thickness and stiffness.**

(A, B) Scheme of the magnetic setup for the cortex pinching experiment: an inverted microscope is associated with two coaxial coils to generate a quasi-homogeneous magnetic field, B, in the sample region (A). Through the application of a magnetic field, beads align, and the cortex is pinched between a bead inside the cell and a bead outside the cell (B). (C) Bright-field image of a WT HoxB8 macrophage, with one internalized and two external magnetic beads aligned by a magnetic field. The red arrows symbolize the pressure that is exerted on the cell cortex. Scale bar: 5 µm. (D) Median of the cortical thickness of 29 WT and 29 ERM-tKO#1 macrophages from three independent experiments were measured by applying a low force (5 mT) between two magnetic beads. Statistics were done using an unpaired t test. Means and SD are shown. (E). Cortical stiffness responses are represented by the tangential elastic modulus at low stress between 150 and 350 Pa. In all, 29 WT and 20 ERM-tKO#1 macrophages from three independent experiments were analyzed. Statistics were done using an unpaired t test. Means and SD are represented. (F) Exemplary force curve from atomic force spectroscopy operated in dynamic tether pulling mode. Tethers break while the cantilever is retracted at a defined velocity with the Z-height increasing constantly. (G) Force-velocity curve from dynamic tether pulling on CTL and ERM-TKO HoxB8 macrophages. Data points are mean tether force f ± SEM at 2, 5, 10, and 30 µm/s pulling velocity. At least 16 cells per condition were analyzed in four independent experiments. (H) Mean and standard deviation of the MCA parameter Alpha obtained from Monte-Carlo-based fitting of the Brochard-Wyart model to the force-velocity data in 5 g (see "Methods" for details). No statistical difference was observed (P value (Z-test): 0.83). (I) Blebbing of Lifeact-mCherry WT and Lifeact-GFP ERM-tKO progenitor cells after incubation with 10% distilled $H_2O$ to induce hypo-osmotic stress. Scale bar: 5 µm. See also Movie EV16. (J) Quantification of the maximum bleb size per blebbing WT and ERM-tKO progenitors extracted from short-time wide-field movies. In total, 38 WT and 99 ERM-tKO#1 cells from three independent experiments were analyzed. Means and SD are shown. (K) Quantification of the retraction speed of blebs from short-time wide-field movies. Overall, 38 WT and 99 ERM-tKO#1 cells from three independent experiments were analyzed. Means and SD are shown. Source data are available online for this figure.

donors before sample collection. The sex of HS is unknown. Human peripheral blood mononuclear cells were isolated from the blood of healthy donors by centrifugation through Ficoll-Paque Plus (Cytiva), resuspended in cold phosphate-buffered saline (PBS) supplemented with 2 mM EDTA, 0.5% heat-inactivated Fetal Calf Serum (FCS) at pH 7.4 and monocytes were magnetically sorted with magnetic microbeads coupled with antibodies directed against CD14 (Miltenyi Biotec #130-050-201). Monocytes were then seeded on glass coverslips at $1.5 \times 10^6$ cells/well in six-well plates in RPMI 1640 (Gibco) without FCS. After 2 h at 37 °C in a humidified 5% $CO_2$ atmosphere, the medium was replaced by RPMI containing 10% FCS and 20 ng/mL of macrophage colony-stimulating factor (M-CSF) (Miltenyi 130096489). For experiments, cells were harvested at day 7 using trypsin-EDTA (Fisher Scientific).

## RNA interference knockdown

For siRNA silencing of all ERMs, 3-day differentiated macrophages were transfected with the HiPerfect system (Qiagen). The mix of HiPerfect and siRNAs was incubated for 15 min at room temperature and cells were then added drop by drop. The following siRNAs were used: human ON-TARGET plus SMART pool siRNA non-targeting control pool (siCtrl Dharmacon #D-001810-10-50); human ON-TARGET plus SMART pool siRNA targeting EZR (siEZR Dharmacon #L-017370-00-0005), RDX (siRDX Dharmacon #L-011762-00-0005) and MSN (siMSN Dharmacon #L-011732-00-0020). The siRNA concentrations were 200 nM for siCtrl and 66.7 nM for siEZR, siRDX and siMSN. The medium was replaced with complete medium containing M-CSF 5 h after transfection to allow complete differentiation for a 5 days.

For moesin silencing, CD14+ human monocytes were transfected with 200 nM siMSN or siCtrl using the HiPerfect system (Qiagen) as described above.

Macrophages were then detached with trypsin-ETDA (Invitrogen) and lysed for Western blot analysis or used for migration assays.

## Murine HoxB8-progenitors culture and HoxB8-macrophages differentiation

Myeloid progenitors were isolated from the bone marrow of a mouse carrying the EF1a-hCas9-IRES-neo transgene in the ROSA26 locus (Tzelepis et al, 2016), and immortalized by transduction with a retrovirus allowing conditional expression of the HoxB8 homeobox gene, as previously described (Accarias et al, 2020). Briefly, bone marrow cells were harvested and hematopoietic progenitor cells were purified by centrifugation using Ficoll-Paque Plus (#17-1440-02). Then, hematopoietic cells were pre-stimulated for 2 days with complete RPMI 1640 medium supplemented with 15% fetal calf serum (FCS), 1% penicillin–streptomycin, 1% glutamine, mouse IL-3, mouse IL-6 and mouse SCF, each at 10 ng/mL (Peprotech). Next, $2 \times 10^5$ cells were seeded on a six-well culture plate in myeloid medium [complete RPMI 1640 medium, supplemented with 20 ng/mL mouse GM-CSF (Miltenyi #130-095-739) and 0.5 µM β-estradiol (Sigma-Aldrich #E2758)] and transduced with 1:5 (vol:vol) Thy1.1-expressing ER-HoxB8 retrovirus supernatant using Lentiblast Premium (OZ Biosciences). After 1 or 2 weeks, only ER-HoxB8-immortalized progenitors continue to proliferate. Based on the expression of Thy1.1 marker at the surface of the immortalized progenitors using flow cytometry, we assessed the clean transduction of the progenitor cells. HoxB8 progenitors were then passaged every 2 days in myeloid medium. To differentiate them in macrophages, progenitors were washed twice in PBS to remove estradiol from the medium, and 500,000 cells were plated in six-well-plate in complete medium containing 20 ng/mL of mM-CSF.

## HoxB8-DC and pre-osteoclast differentiation

To differentiate HoxB8 progenitors into dendritic-like cells, cells were washed twice in PBS to remove estradiol from the medium, and 500,000 cells were plated in a Petri dish (Fisher scientific #10470613) in 10 mL of complete medium containing 40 ng/mL of mouse GM-CSF. On day 3 of culture, 10 mL of medium + GM-CSF was added. Half of the medium was replaced on day 6. On day 8, non-adherent cells were collected from the supernatant and used for the experiments (Leithner et al, 2018). Differentiation status was assessed by immunostaining and flow cytometry. GM-CSF was kept in the culture medium during migration experiments.

To differentiate HoxB8 progenitors into pre-osteoclasts, 400,000 cells were plated in six-well-plates in complete medium containing 30 ng/mL of mouse M-CSF and 50 ng/mL RANKL (Miltenyi #130-094-646) for 3 days. The cells were then detached using Accutase

(StemCell #07920) and used for the experiments (Vérollet et al, 2013). M-CSF and RANKL were kept in the culture medium during migration experiments.

## Flow cytometry

After a 5 min centrifugation at $320 \times g$, cell pellets were resuspended in cold staining buffer (1% FBS, 2 mM EDTA in PBS) and incubated on ice for 20 min. Cells were then stained with APC/Cy7 Ly6C (Biolegend #128025), Brilliant Violet® 605 F4/80 (Biolegend #123133), PE CD11b (Biolegend #101207) and APC CD11c (Biolegend #117310) antibodies or their isotype controls, all at 1:200 dilution in staining buffer, and incubated on ice for 30 min. Cells were then washed twice, resuspended in PBS and analyzed by using a BD LSR Fortessa (BD Biosciences) and the associated BD FACSDiva software. Data were processed using the FlowJo_V10 software. Live cells were first selected according to their Forward and Size Scatter (FSC and SSC) properties, followed by doublet exclusion. The Ly6c- population was quantified and gated to analyze the remaining markers.

## Transfection, transduction, and sequence analysis for the creation of single and triple-knockout in murine HoxB8 progenitors

SgRNAs used for knocking out ERM proteins were EZR: (5′-CT ACCCCGAAGACGTGGCCG-3′), RDX: (5′-GCCATCCAGCCC AATACAAC-3′), and MSN: (5′-TATGCCGTCCAGTCTAAGTA-3′). We targeted luciferase as a control, with LUC sgRNA: (5′-GG CGCGGTCGGTAAAGTTGT-3′). For single knockouts, sgRNAs were introduced into pLenti-sgRNA backbone (Addgene #71409) with BsmBI digestion (New Englands Biolabs #R0739) and T4 DNA ligase (Fischer Scientific EL011L). For the triple ERM-knockout, all three sgRNAs were cloned on a same plasmid by golden gate assembly (Kabadi et al, 2014). Briefly, phU6-gRNA (#53188) intermediary plasmid was used to clone EZR gRNA, pmU6-gRNA (#53187) to clone RDX and LUC gRNAs, #53189 ph7SK-gRNA for MSN, and a mock (5′-TTTTTTTTTTT-3′) sequence was cloned into #53186 phH1-gRNA, and the three plasmids previously mentioned. All intermediary plasmids containing respective sgRNA were then cloned into the final pLV hUbC-Cas9-T2A-GFP plasmid (Addgene #53190), where GFP was replaced by puromycin resistance.

Then, HoxB8 progenitors were knocked out for ezrin, radixin or moesin using CRISPR–Cas9. Briefly, HEK293T cells were co-transfected with pMDL (Addgene #12251), pREV (Addgene #12253), pVSV-G (Addgene #12259) and specific sgRNA-cloned plasmids using Lipofectamine 3000 and OptiMEM according to manufacturer's guidelines to generate lentiviral particles for transduction into Cas9-expressing HoxB8 progenitors. Then, $2 \times 10^5$ HoxB8 progenitors were transduced with viral particles and Lentiblast Premium (OZ Biosciences). After 24 h, transduced cells were selected with 10 µg/mL of puromycin (Invivogen) for 2 days and validated by immunoblotting. For the triple ERM-KO only, a clonal amplification was performed by single-cell sorting, and clones were screened for ezrin, radixin, and moesin absence of expression by flow cytometry. Immunoblotting with respective ERM antibodies confirmed the depletion of the three ERM proteins in three independent clones (tKO#1, tKO#2, tKO#3). In addition,

total genomic DNA of clonal populations was extracted using TRIzol Reagent (ThermoFisher). Genomic sites encompassing targeted guide regions were amplified by PCR using GoTaq polymerase (Promega) and sequenced, in order to analyze the genomic point mutations acquired by each clone on the three E, R, M genes.

## Western blot

The following rabbit antibodies were purchased from Cell Signaling: anti-ezrin (#3145S, 1/1000), anti-moesin (#3150, 1/1000), anti-radixin (#2636S, 1/1000), anti-pan-ERM (#3142P, WB 1/1000). Rabbit anti-phospho-ERM (1/5000) (Roubinet et al, 2011) and rabbit anti-actin (Sigma-Aldrich A5060, 1/10,000) were also used. Cells were lysed by the addition of boiling 2× Laemmli buffer containing phosphatase inhibitors (5 mM sodium orthovanadate, 20 mM sodium fluoride and 25 mM β-glycerophosphate). Proteins were subjected to electrophoresis in 8% SDS–PAGE gels, and transferred onto a nitrocellulose membrane. Membranes were saturated with 3% BSA in TBS-T (50 mM Tris, pH 7.2; 150 mM NaCl; and 0.1% Tween 20) for 30 min, and incubated with primary antibodies overnight at 4 °C. Then, primary antibodies were revealed using HRP-coupled secondary anti-mouse (Sigma) or anti-rabbit antibodies (Cell Signaling) for 1 h. Finally, HRP activity was revealed using an electrochemiluminescence kit (Amersham) according to the manufacturer's instructions. All quantifications were normalized against actin expression.

## In vitro migration assays

### 2D random migration

In total, $2-6 \times 10^3$ HoxB8 macrophages were harvested and plated on glass-based 96-wells within complete medium. 2D random migration was recorded every 3 min for 16 h with the ×10 objective of a wide-field EVOS M7000 (Invitrogen) in an environmental chamber maintaining the temperature at 37 °C and 5% $CO_2$. After cell segmentation, tracking of macrophages was done using the automated Trackmate plugin of ImageJ with the LogDetector and Simple LAP tracker. Only the tracks lasting more than 1 h were kept for the analysis. In addition to the median speed, the confinement ratio was defined as the ratio of the net distance on the total distance traveled by the cell.

### 2D chemotaxis toward complement 5a (C5a)

In total, 25,000 HoxB8 macrophages were seeded in the center part of a µ chemotaxis slide (Ibidi) and let to adhere at 37 °C and 5% $CO_2$ for 10 min. The peripheral chambers were filled with 65 µL of complete medium containing 20 ng/mL murine M-CSF. A C5a gradient (R&D systems #2150C5025) was then generated by adding 30 µL of medium containing 60 nM murine recombinant C5a protein to the left part of the slide. Cells were imaged as in the 2D random assay every 3 min for 16 h. The FastTrackAI artificial Intelligence performed cell tracking, and tracks were filtered using the Chemotaxis and Migration tool (Ibidi, free download from http://www.ibidi.de/applications/ap_chemo.html).

### 3D transwell migration in Matrigel and collagen I

For 3D migration assays, 24-transwells (8-µm pores) were loaded with either 100 µl of 10 mg/mL Matrigel™ (Corning) or 110 µl of

2.15 mg/mL fibrillar collagen I (Nutragen), mixed with MEM 10×, H2O, and buffered with bicarbonate solution (Van Goethem et al, 2010). Matrices were allowed to polymerize for 30 min at 37 °C, and rehydrated overnight with RPMI 1640 without FCS. The lower chamber was then filled with RPMI 1640 containing 10% FCS and 20 ng/mL mouse M-CSF, and the upper chamber with RPMI 1640 containing 2% FCS and 20 ng/mL mouse M-CSF. Overall, $5 \times 10^4$ macrophages were serum starved for 3 h and seeded in the upper chamber. Each experiment was performed in triplicate. After 24 h, 48 h or 72 h of migration, z-series of images were acquired at the surface and inside of the matrices with 30-μm intervals. Acquisition and quantification of cell migration was performed using the motorized stage of an inverted video microscope (Leica DMIRB, Leica Microsystems, Deerfield, IL) and the cell counter plugin of the ImageJ software as described previously (Van Goethem et al, 2010). The percentage of migration was obtained after counting the number of cells within the matrix and dividing by the total number of cells. The distribution of the cells into the matrix is represented by plotting the migration distance of each cell.

## Ex vivo infiltration of mouse macrophages in tumor and ear derma tissue explants

### Ethics

All experiments on animals were performed according to animal protocols approved by the Animal Care and Use Committee of the Institute of Pharmacology and Structural Biology (APAFIS No. 201609161058531).

### Preparation of tumor explants

Overall, $1 \times 10^6$ LPB fibrosarcoma cell were injected subcutaneously in the flank of C57BL/6 mice to induce a fibrosarcoma (Gui et al, 2018). In all, 1-cm³ tumors were resected and embedded in 3% low-gelling agarose (Sigma-Aldrich #A701) prepared in PBS. In all, 500 μm-thick slices were obtained with the Krumdieck tissue slicer (TSE Systems) filled with ice-cold PBS (Life Technologies) (Gui et al, 2018).

### Preparation of ear derma tissue

Depilated ears from sacrificed C57BL/6 mice were resected. Ear halves were separated under stereomicroscope (Leica microsystems) lengthwise to expose the ear derma.

### Ex vivo infiltration assays of mouse macrophages in tissue explants

Tumor slice and ear derma explants were then cultured on a 30-mm cell culture insert featuring a hydrophilic PTFE membrane (0.4 μm pore size, Merck Millipore #PICM0RG50) placed inside 6-well plates containing 1.1 mL of RPMI. A 4 mm-diameter stainless-steel washer was then placed on top of each tissue slice to create a well for macrophage seeding. The same day, HoxB8-macrophages CTL and TKO-ERM were differentially labeled with 5 μM Green CellTracker CMFDA (Invitrogen #C7025), or Red CellTracker CMPTX (Invitrogen #C34552) for 30 min at 37 °C. Macrophages were harvested and a 1:1 mix of CTL and TKO-ERM macrophages was seeded ($1.5 \times 10^4$ cells/mm²) on top of tissue explants and incubated in a 37 °C, 5% $CO_2$ environment for 72 h. After 16 h of coculture, the washer was removed. Culture medium (RPMI 1640, Gibco) was replaced daily for 3 days before overnight paraformaldehyde 4% (EMS #15714) tissue fixation at 4 °C and

embedded in paraffin. The green and red cell trackers were exchanged between experiments in order to verify the absence of effect due to staining.

In all, 5 μm-thick ear or tumor paraffin sections were stained with hematoxylin and eosin (HE) or immunostained by immuno-histofluorescence. To limit tissue autofluorescence, CMFDA staining of HoxB8 infiltrate in tissue, was stained by anti-fluorescein (1:100, Invitrogen #A889) by overnight incubation at 4 °C. The sections were incubated at room temperature with goat anti-rabbit AlexaFluor 647 (1:200, Invitrogen #A21246) for 2 h and nuclei were visualized with DAPI (Sigma D9542). The CMPTX cell tracker fluorescence was directly acquired. All images were acquired using a Zeiss Axio Imager M2 using a X20/0.8 Plan Apochromat objective (Zeiss). Images were acquired with an ORCA-flash4.0 LT (Hamamatsu) camera and processed using Zen software (Zeiss).

For each image, the percentage of infiltrated HoxB8-WT or TKO macrophages, calculated with a threshold of 10 μm under the tissue surface, was compared to the total number of macrophages infiltrated in the tissue. Quantifications were done using ImageJ "*Threshold and Analyze particles*" tool to select CMFDA or CMPTX positive cells, and the measure of migration distance was enabled using the "*Segmented line*" and the "*Straighten*" tools to linearize tissue surface. For ear tissue infiltration, we performed the analysis on four ear halves from four mice, taking 1–4 sections per ear half. For analysis of macrophage tumor infiltration, we used 6 ex vivo 500-μm slices from three tumors, taking 7–9 sections per ex vivo slice.

## In vivo analysis of adhesion to vascular endothelium with wide-field intravital microscopy

All experiments on animals were performed according to animal protocols approved by the Animal Care and Use Committee of the Institute of Pharmacology and Structural Biology (pilot procedure 2021-PP-VER-01).

C57Bl6Jj female mice were subcutaneous injected 100 μL PBS with $5 \times 10^5$ MCA Prog (9609) fibrosarcoma tumor cell to induce a fast-growing fibrosarcoma (O'Sullivan et al, 2012). Intravital microscopy was performed at day 8 to 10 before the tumor reach 1 cm³.

HoxB8 progenitors were directed towards monocyte/macrophage differentiation using a 1-day treatment with 20 ng/mL mouse M-CSF. CTL and TKO-ERM cells were harvested and differentially labeled with 10 μM Green CellTracker CMFDA (Invitrogen #C7025) and 10 μM Red CellTracker CMPTX (Invitrogen #C34552) for 20 min. Green and red cell trackers were exchanged between experiments. A 1:1 mix of CTL:TKO-ERM cells was resuspended in physiological serum.

Recipient mice were anesthetized by intraperitoneal injection of 1 mg/mL xylazine and 5 mg/mL ketamine per kg/mouse, and placed on a customized stage for securing animals and immobilizing the subcutaneous tumor. A heating pad with temperature feedback to an mTCII micro Temperature Controller (Cell MicroControls, Norflok, VA, USA) was used to maintain the temperature of animals. The right femoral artery was catheterized for retrograde injection of labeled cells, and tumor was exposed for intravital microscopy as previously described (Moussion and Girard, 2011; von Andrian, 1996). Recipient mice were then transferred to a customized intravital video microscopy setup (DM6-FS, Leica

Microsystems SAS, Nanterre, France) equipped with water immersion objectives (HCX APO; Leica Microsystems) and an image splitting optics W-VIEW GEMINI (Hamamatsu Photonics, Massy, France) allowing simultaneous image acquisition of dual-wavelength images (512/25 and 630/92 nm for green and red cell tracker respectively) onto a single camera. Two color fluorescent events in the microcirculation of tumor/draining lymph node were visualized and recorded at 33 frames/seconde by an OrcaFlash LT4.0+ camera and the HCImage software (Hamamatsu Photonics) at multiple positions of the tumor. After the combination of green and red images on a single image, rolling fractions, i.e., the percentage of rolling cells in the total flux of cells in each blood vessel; and sticking fractions, i.e., the percentage of rolling cells that firmly adhered for a minimum of 30 s, were calculated for all blood vessels imaged in tumor.

## Immunofluorescence and cell morphological analysis

For analysis of the morphology of TKO-ERM macrophages, HoxB8-macrophages CTL and TKO-ERM were differentially labeled with 10 µM Blue CellTracker CMAC (Invitrogen #C2110) or 10 µM Red CellTracker CMPTX (Invitrogen #C34552) for 30 min. Blue and red cell trackers were exchanged between independent experiments. Macrophages were harvested, and a 1:1 mix of CTL and TKO-ERM macrophages were plated on glass coverslips for 1 h. Cells were fixed for 10 min in 3.7% paraformaldehyde solution containing 15 mM sucrose in PBS at room temperature, and unreacted aldehyde functions were quenched with 50 mM NH4Cl in PBS for 5 min at RT. Then cells were permeabilized for 10 min with PBS/0.3% Triton and blocked with PBS/0.3% Triton/1% BSA. Samples were incubated with green AlexaFluor 488-conjugated phalloidin (Invitrogen) for 30 min, then washed and mounted. Mosaic images were acquired using a Zeiss Axio Imager M2 using a X100/1.4 Plan Apochromat objective (Zeiss). Cells were semi-automatically segmented to measure their area and circularity based on their actin staining. WT and TKO-ERM cells were differentiated based on their respective cell tracker, which were inversed between experiments to ensure they do not have an effect. Then, podosomes were detected using the "Find Maxima" function of the ImageJ software. Filopodia of 30 cells for each condition were manually traced to get their length and number per cell.

## Phagocytosis assays

Polystyrene beads (Fluoresbrite yellow-green, Polyscience, 3-µm diameter) were coated with ovalbumin (1 mg/ml in PBS) for 1 h at room temperature and washed with PBS containing 1% BSA. To obtain IgG-opsonized beads (IgG beads), ovalbumin-coated beads (OVA beads) were then incubated with anti-ovalbumin antibodies for an hour (1:10, room temperature) and washed three times.

Particles were added to macrophages at 4 °C at several multiplicities of infection (MOI) to obtain equivalent rates of phagocytosis. Ovalbumin-coated beads were used at MOI 24:1 (beads:macrophages) and IgG beads at MOI 4:1 with macrophages plated on coverslips. Phagocytosis was synchronized with centrifugation at $170 \times g$ for 1 min. Unbound beads were washed off with culture medium and cells were then incubated at 37 °C and fixed at the indicated time points (see "Immunofluorescence methods").

Prior to permeabilization, IgG-coated beads that remained outside the cells were distinguished from ingested beads using Goat anti-Rabbit AF647 antibodies (Invitrogen), while OVA beads, were discriminated with anti-ovalbumin antibodies (1:150 dilution, Millipore) revealed by Goat anti-Rabbit AF647 antibodies. Cells were then washed, permeabilized before being stained with DAPI and fluorescent phalloidin for 30 min at RT. After washing, samples were then mounted with DAKO. Statistical analyses were performed using regular two-way ANOVA followed by Boneferroni's comparison test using Prism 5 (GraphPad).

## Live structural illumination microscopy (SIM) imaging of ezrin, radixin, and moesin transfected in human macrophages

For live imaging of ERM proteins shown in Movies EV1–6, primary human macrophages were detached using trypsin-EDTA and transfected with Lifeact-mCherry and either ezrin-GFP, radixin-GFP or moesin-GFP plasmids 4 h before observation using a Neon® MP5000 electroporation system (Invitrogen) with the following parameters: two 1000 V, 40 ms pulses, with 0.5 µg DNA each for $2 \times 10^5$ cells. Ezrin-GFP from Leguay et al, 2022, Radixin-GFP and Moesin-GFP were obtained by subcloning PCR amplified Ezrin, Radixin, and Moesin, respectively, into pEGFP-N1 vector (Clontech). All PCRs were performed using Phusion High-Fidelity DNA Polymerase (New England Biolabs).

SIM live images of double-transfected cells were acquired every 15 s using a Plan-Apochromat 63x/1.4 Oil DIC M27 on an Elyra 7 microscope (Zeiss) and 129 nm-interspaced Z-stacks were also acquired. Super-resolution reconstruction was done with Zen Black (3.0 SR) SIM algorithm 3D leap processing with autosharpness. Chromatic aberrations were corrected with a channel alignment calculated from a multicolor subresolutive beads acquisition from the same half-day.

## Scanning electron microscopy

HoxB8 progenitors and macrophages were fixed using 0.1 M sodium cacodylate buffer (pH 7.4) supplemented with 2.5% (v/v) glutaraldehyde. After post-fixation in 1% osmium tetraoxide (in 0.2 mol/L cacodylate buffer), cells were dehydrated in a series of increasing ethanol concentrations and critical point dried using carbon dioxide. After coating with gold, cells were examined with a JEOL JSM-6700F scanning electron microscope. For image analysis, protrusions of each HoxB8-progenitor cells present in a square of $4 \, \mu m^2$ were manually counted. Lamellipodia of HoxB8 macrophages were manually detoured and expressed as a percentage of the cell perimeter.

## Live random illumination microscopy of actin structures

For live imaging of the actin remodeling in 2D and 3D matrices, shown in Movies EV7, 14, and 15, HoxB8-WT and ERM-TKO (tKO#1) progenitors were stably transduced with Lifeact-GFP plasmid-containing virus. Cells were sorted by flow cytometry according to their GFP expression. Subsequent differentiated macrophages were detached using PBS-EDTA and plated on Ibidi wells (eight wells Ibidi #80826), either directly in 2D bare glass, or mixed with 100% Matrigel, or 2.15 mg/mL collagen I at $2 \times 10^6$ cells per mL, and allowed to polymerize for 30 min in the incubator before adding complete medium. Random Illumination Microscopy was then performed using

a homemade system (Mangeat et al, 2021). Images were acquired every 10 s for 5 min using an inverted microscope (TEi Nikon) equipped with a ×100 magnification, 1.49 N.A. objective (CFI SR APO 100XH ON 1.49 NIKON) and an sCMOS camera (ORCA-Flash4.0 LT, Hamamatsu). Fast diode lasers (Oxxius) with respective wavelengths 488 nm (LBX-488-200-CSB) and 561 nm (LMX-561L-200-COL) were collimated using a fiber collimator (RGBV Fiber Collimators 60FC Sukhamburg) to produce TEM00 2.2 mm-diameter beam. The polarization beam was rotated with an angle of 5° before hitting a X4 Beam Expander beam (GBE04-A) and produced a 8.8 mm TEM00 beam. A fast spatial light phase binary modulator (QXGA fourth dimensions) was conjugated to the image plane to create speckle random illumination. Image reconstruction was then performed as detailed in (Mangeat et al, 2021) and at https://github.com/teamRIM/tutoRIM.

Analysis of podosome maintenance in 2D was performed using ImageJ. Briefly, we quantified the number of podosomes appearing, maintained, or disappearing between four images at 1-min intervals (Appendix Fig. S2). Analysis of ruffle retraction in 3D collagen I was performed as follows. The ruffle dynamics were analyzed in videos of 300 s with a 10 s frequency: using ImageJ, we measured the length of the protrusion at its maximal elongation, and the time until the protrusion disappeared.

## Measure of cortex thickness and dynamics

### Beads preparation
Superparamagnetic microbeads M-450 (Dynabeads) were prepared in two solutions. For both, 60 μL of stock solution was washed three times with PBS using a magnet to retain the beads in the pellet between each rinsing. They were then resuspended in 200 μL of complete medium (DMEM + 10% FBS) for the bead solution n°1, and in 200 μL of 0.5 mg/mL PLL-g-PEG (Peg 2 kDa, SuSos) in HEPES for the bead solution n°2. Both were placed on a wheel for 8 h to get coated by their respective medium. In the following methods, beads from the first solution were incubated with cells to be ingested, while those from the second solution were intended to be less likely to be taken.

### Magnetic setup
As described in (Laplaud et al, 2021), an Axio A1 inverted microscope (Carl Zeiss, Germany) is used as the base for the Magnetic Pincher setup. Imaged are produced using an oil-immersion ×100 objective [1.4 numerical aperture] mounted on a piezo-controlled translator (Physik Instrumente) and recorded with an Orca Flash4 camera (Hamamatsu Photonics). Two coaxial coils (SBEA) with mu metal core (length, 40 mm; diameter, 26–88 mm; 750 spires) are placed in Helmholtz configuration in a way to generate a quasi-homogeneous magnetic field in the sample region. They are powered by a bipolar operational power supply amplifier 6 A/36 V (Kepco) controlled by a data acquisition module (National Instruments). The maximum field generated is 54 mT with a gradient less than 0.1 mT mm$^{-1}$ over the sample. During an experiment, the self-organization of beads is first triggered with a constant field of 5 mT before the acquisition of time-lapse images. The sample is maintained at 37 °C using the Box and the Cube from Life Imaging Systems. The setup is controlled via a data acquisition module by a custom LabVIEW interface that ensures the synchronicity between piezo position, magnetic field imposition, and image acquisition.

### Pinching experiment
CTL and ERM-TKO HoxB8 macrophages were plated on glass in Petri dishes and in each of them 1 μL of the bead solution n°1 was added ($1.5 \times 10^5$ beads) and incubated for 30 min so that the cells would take up the beads. Then 2 μL of bead solution n°2 was added, and the medium was supplemented with 20 mM HEPES buffer, to maintain the pH of the medium during the experiment without $CO_2$ control. For each dish, cells with their cortex pinched between beads were imaged with the previously described magnetic setup. In each film acquisition, the nominal magnetic field exerted by the coils was 5 mT (100pN), every 10 s the field was lowered to 1 mT (31 pN), then increased to 54 mT (1150 pN) in 1.5 s, and then brought back to 5 mT (values between brackets are the corresponding typical force values). This series of compression-relaxation was repeated 6–10 times per cell, and an epifluorescence image of the LifeAct-GFP signal was taken between each cycle. In addition to the images, the precise time and magnetic field corresponding to each image was recorded.

### Data analysis
Using Fiji's plugin "Analyze Particle", the center of all beads on each image of time lapse was detected with a subpixel resolution (Laplaud et al, 2021). Using a homemade tracking algorithm written in Python (https://github.com/jvermeil-biophys/CortExplore_PublicVersion) the trajectories in 3D of the centers of the two beads pinching the cortex were detected, and the thickness of the pinched cortex was computed. The pinching force was also computed knowing the distance between the beads, the external magnetic field, and the magnetization function of the beads. This analysis was complemented by a visual inspection of the LifeAct-GFP images of the cells and compressions occurring while a phagocytosis process had begun were removed from the dataset.

To characterize the cortex thickness, the metric we chose is the median of the cortical thickness at 5 mT (nominal field), which corresponds to a typical force of $100 + / -18$ pN.

To compute the elastic modulus at low stress, we computed the strain-stress curve corresponding to each compression using a model developed by R. S. Chadwick for this specific geometry (Chadwick, 2002). Then we fitted the slope of these strain-stress curves between 150 and 350 Pa.

## Atomic force spectroscopy

CTL and ERM-TKO HoxB8 macrophages were plated into a 35 mm Fluorodish (WPI) 24 h before the experiment. 30 min prior to the AFM experiment, the serum concentration in the medium was reduced to 2% FBS.

qp-SCONT cantilevers (Nanosensors) were mounted on a CellHesion 200 AFM (Bruker), connected to an Eclipse Ti inverted light microscope (Nikon). Cantilevers were calibrated using the contact-based approach, followed by coating with 4 mg/ml Concanavalin A (Sigma) for 1 h at 37 °C. Cantilevers were washed with 1× PBS before the measurements. MCA (Membrane-to-Cortex Attachment) was estimated using dynamic tether pulling as follows: Approach velocity was set to 0.5 μm/s, with a contact force of 200 pN, and contact time was varied between 100 ms to 10 s, aiming at maximizing the probability to extrude single tethers. The cantilever was then retracted for 80 μm at a velocity of 2, 5, 10, or 30 μm/s. Tether force at the moment of tether breakage was recorded at a sampling rate of 2000 Hz. Resulting force curves were analyzed using the JPK Data Processing Software and the resulting

force-velocity data was fitted via Monte-Carlo Simulations ($n = \sim5000$) to the Brochard-Wyart model (Brochard-Wyart et al, 2006). This allows estimation of an MCA parameter Alpha, which is proportional to the density of binders (i.e., the active MCA molecules) and the emerging effective viscosity. Measurements were run at 37 °C with 5% $CO_2$ and samples were used no longer than 1 h for data acquisition. More details on the method can be found in (Bergert and Diz-Muñoz, 2023).

## Statistical analysis

Statistical analysis was done using GraphPad Prism 9.0 (GraphPad Software Inc.). For western blot quantifications, the expression of each protein in human or HoxB8 macrophages was compared to monocytes or Hox progenitors, two by two, respectively, using a ratio paired two-tailed *t* test. For all migration experiments including 3D migration percentage, distance, and the velocity, directionality, FMIx, and confinement ratio of 2D migrations; the means of each three independent experiments was compared to WT condition using either paired *t* test for comparison of two conditions, or RM ANOVA one-way for more than two conditions.

For morphological experiments based on SEM, and actin immunofluorescence analysis, data were compared between TKO cells and control cells using the two-tailed Mann–Whitney test. In all cases * correspond to $P < 0.05$.

The analyses were carried out blind.

# Data availability

The source data of this paper are collected in the following database record: biostudies:S-SCDT-10_1038-S44318-024-00173-7.

# Peer review information

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

## Acknowledgements

This work benefited from the assistance of Isabelle Fourqueaux (CMEAB), Emmanuelle Naser and Eve Pitot (IPBS) from the imaging and cytometry facility TRI, member of the national infrastructure France-BioImaging infrastructure supported by the French National Research Agency (ANR-10-INBS-04) and Etienne Meunier for providing access to wide-field EVOS microscope. The authors thank the Wellcome Sanger Institute for providing a mouse line expressing Cas9 nuclease and Tim Lammermann, Paul Mangeat, Eva Kiermaier, Emmanuelle Planus, Pablo Vargas, Christel Lutz, Geanncarlo Lugo-Villarino, François Payre and Damien Ramel for helpful discussions. This work was supported by the Université Toulouse III - Paul Sabatier (UT3), ITMO Cancer of Aviesan on funds managed by Inserm (RP), the CIHR (PJT162109) (SC), Globalink-MITACS (PV), l'Agence Nationale de la Recherche (ANR-21-CE13-0048) and the Institut Pierre-Gilles de Gennes-IPGG (Equipement d'Excellence, "Investissements d'avenir," program ANR-10-EQPX-34 and Laboratoire d'Excellence, "Investissements d'avenir" program ANR-10-IDEX-0001-02 PSL and ANR-10-LABX-31) (JV and ODR) and the European Molecular Biology Laboratory (EMBL) (MB and ADM).

## Author contributions

**Perrine Verdys**: Data curation; Formal analysis; Investigation; Visualization; Methodology; Writing—original draft; Writing—review and editing. **Javier Rey Barroso**: Data curation; Formal analysis; Investigation; Visualization; Methodology; Writing—original draft; Writing—review and editing. **Adeline Girel**: Data curation; Formal analysis; Investigation. **Joseph Vermeil**: Data curation; Formal analysis; Validation; Investigation; Visualization; Methodology; Writing—original draft. **Martin Bergert**: Data curation; Formal analysis; Validation; Investigation; Visualization; Methodology; Writing—original draft. **Thibaut Sanchez**: Investigation; Methodology; Writing—original draft. **Arnaud Métais**: Formal analysis; Validation; Investigation; Visualization; Methodology. **Thomas Mangeat**: Formal analysis; Investigation; Visualization; Methodology. **Elisabeth Bellard**: Formal analysis; Validation; Investigation; Visualization; Methodology. **Claire Bigot**: Formal analysis; Investigation; Visualization; Methodology; Writing—review and editing. **Catherine Astarie-Dequeker**: Formal analysis; Validation; Investigation; Visualization. **Arnaud Labrousse**: Formal analysis. **Jean-Philippe Girard**: Resources; Supervision; Validation; Writing—original draft; Writing—review and editing. **Isabelle Maridonneau-Parini**: Supervision; Funding acquisition. **Christel Vérollet**: Funding acquisition; Validation; Writing—original draft. **Frédéric Lagarrigue**: Resources; Supervision; Validation; Methodology. **Alba Diz-Muñoz**: Supervision; Funding acquisition; Validation; Writing—original draft. **Julien Heuvingh**: Supervision; Funding acquisition; Validation; Writing—original draft. **Matthieu Piel**: Supervision; Funding acquisition; Validation; Writing—original draft. **Olivia du Roure**: Supervision; Funding acquisition; Validation; Writing—original draft. **Véronique Le Cabec**: Conceptualization; Supervision; Validation; Writing—original draft; Writing—review and editing. **Sébastien Carréno** Conceptualization; Supervision; Funding acquisition; Validation; Visualization; Writing—original draft; Project administration; Writing—review and editing. **Renaud Poincloux**: Conceptualization; Formal analysis; Supervision; Funding acquisition; Validation; Visualization; Writing—original draft; Project administration; Writing—review and editing.

Source data underlying figure panels in this paper may have individual authorship assigned. Where available, figure panel/source data authorship is listed in the following database record: biostudies:S-SCDT-10_1038-S44318-024-00173-7.

## Disclosure and competing interests statement

The authors declare no competing interests.

# Expanded View Figures

**Figure EV1.  Localization of Ezrin, Radixin and Moesin proteins in human macrophages.**

(**A–C**) Representative SIM images of HMDM co-transfected with ezrin-GFP (green) (**A**), radixin-GFP (green) (**B**) or moesin-GFP (green) (**C**) and Lifeact-mCherry (magenta) at the basal membrane, showing podosomes ($z = 0\ \mu m$) and at $3\ \mu m$ above the basal membrane, showing membrane ruffles (left panels). Scale bars: $10\ \mu m$, enlarged view: $1\ \mu m$. Intensity profiles along the dotted line from both enlarged view of left panels, crossing podosomes ($z = 0\ \mu m$) and membrane ruffles ($z = 3\ \mu m$) (right panels). Also see z-stack Movies EV1, 2 and 3. The fluorescence levels were adjusted in the same way in order to compare the intensity at the base of the cells to the upper planes. Note that ERM are mainly accumulated in the upper ruffles, compared to the basal plasma membrane and that only Ezrin slightly accumulate around podosome cores. (**D**) Enlarged view of ruffle dynamics from SIM images of HMDM co-transfected with ezrin-GFP (left panel), radixin-GFP (middle panel) or moesin-GFP (right panel) (green) and Lifeact-mCherry (magenta). Scale bars: $1\ \mu m$. ERM-GFP (green) or actin (magenta) intensity profiles along the dotted line are plotted below. Note that peripheral ruffles are enriched in F-actin, whereas ERM are present in both peripheral and central ruffles. Also see time-lapse Movies EV4, 5 and 6.

▶

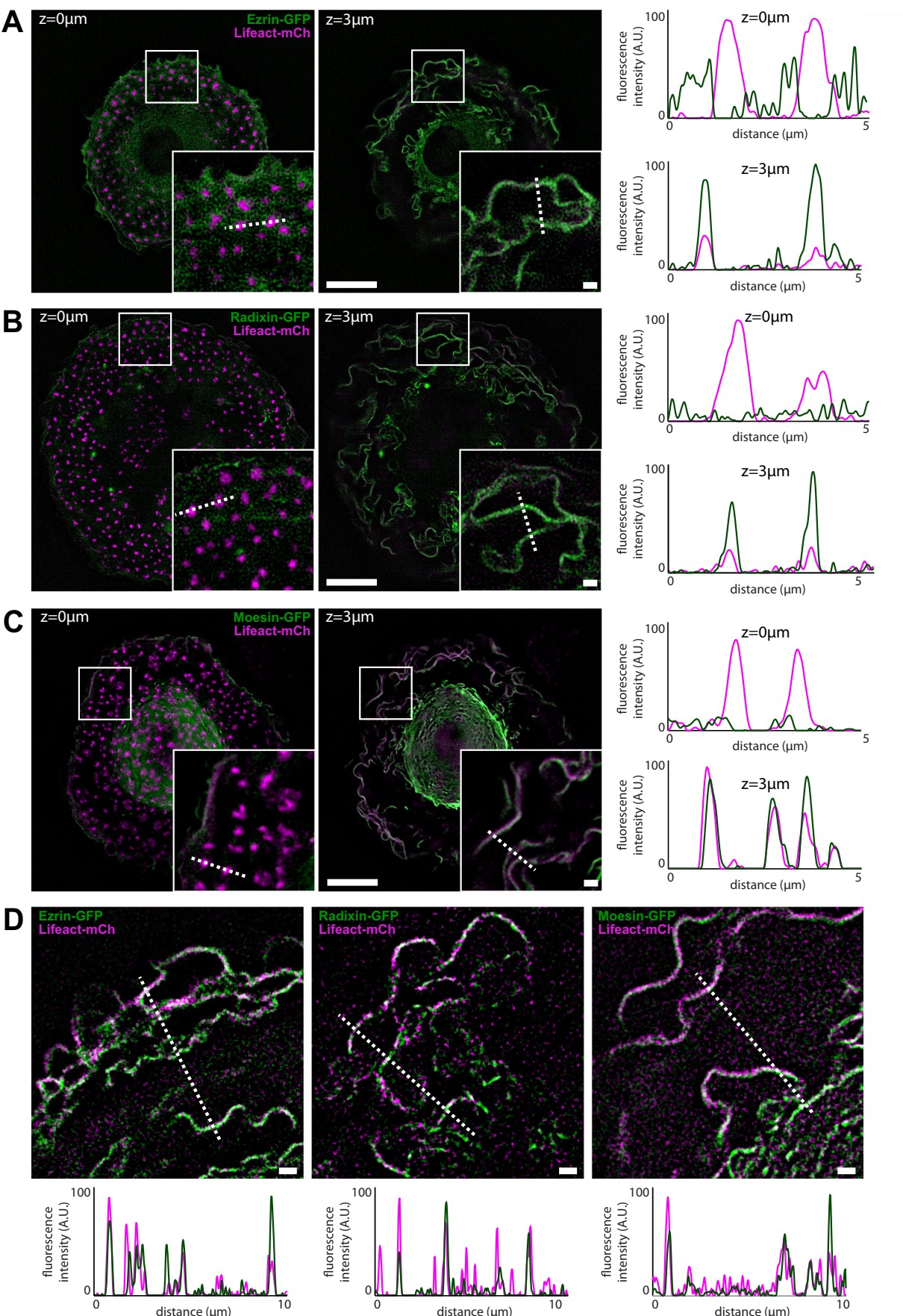

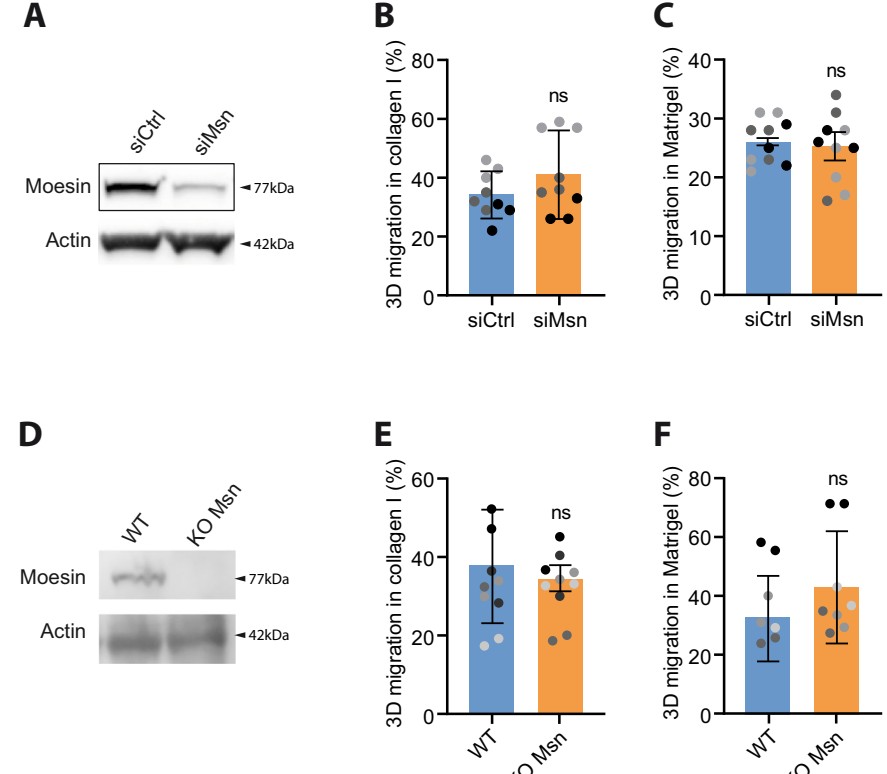

**Figure EV2. Moesin siRNA and KO does not affect macrophage 3D migration.**

(**A–C**) Depletion of Moesin in human macrophages by siRNA. (**A**) Moesin expression level of HMDM treated with siCtrl or siMoesin (siMSN) is representative of 3 independent donors. (**B, C**) Percentages of migration of siRNA-treated HMDM inside collagen I (**B**) and Matrigel (**C**) are represented as follows: the technical replicates (dot) of 3 independent experiments (highlighted by different gray colors) are represented. The mean (bar) and SD from the 3 independent experiments are shown. Statistical analysis was done on the mean per experiment using a paired two-tailed *t* test. (**D–F**) Moesin KO in mouse macrophages. (**D**) Moesin expression level in WT or Moesin KO mouse macrophages, differentiated in macrophage directly after KO induction to avoid compensations, is representative of 3 independent KO. (**E, F**) Percentages of migration inside collagen I (**E**) and Matrigel (**F**) are represented as follows: the technical replicates (dot) of 4 (collagen I) and 5 (Matrigel) independent experiments (highlighted by different gray colors) are represented. The mean (bar) and SD from the independent experiments are shown. Statistical analysis was done on the mean per experiment using a paired two-tailed t test.

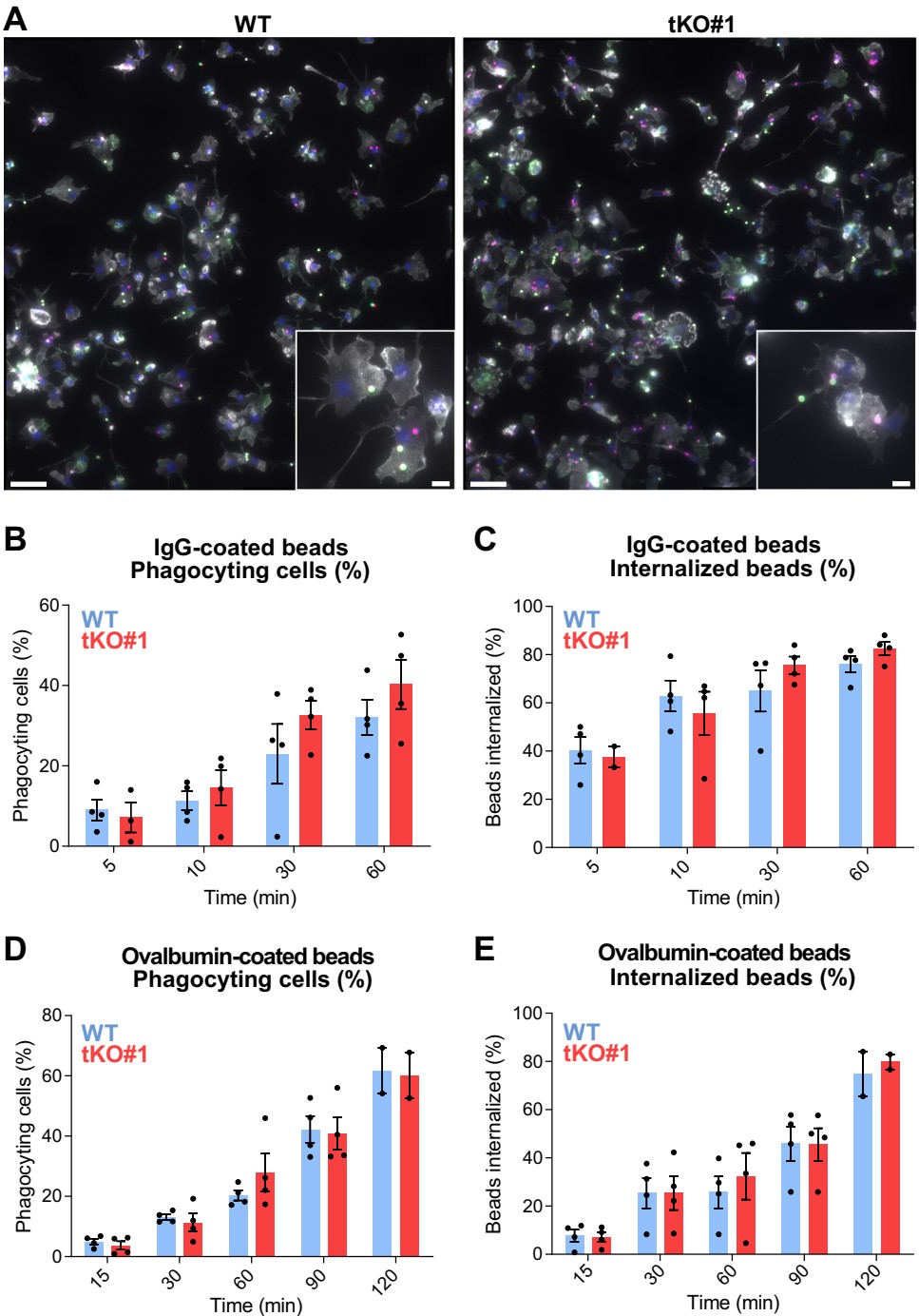

**Figure EV3. Phagocytosis by WT and ERM-tKO macrophages.**

HoxB8 macrophages were exposed to fluorescent IgG beads or OVA beads. Centrifugation was used to synchronize phagocytosis and cells were fixed at the indicated times (5 to 120 min). (**A**) Representative images of fluorescence microscopy of macrophages exposed to OVA beads for 60 min are shown. Beads that remained outside the cells were distinguished from ingested beads using anti-ovalbumin antibodies and TRITC-coupled secondary antibodies. Beads inside cells are magenta, beads outside cells are green, F-actin is shown in white and nuclei in blue. 3 × 3 tile images were stitched together with Zen software. Scale bars: 50 μm or 10 μm for zooms. (**B–E**) The percentages of phagocyting cells (**B, D**) and percentages of fully internalized beads (**C, E**) were quantified for both IgG beads and OVA beads. Results are expressed as mean +/− SD of at least 2600 cells/time point from 4 independent experiments and analyzed with two-way ANOVA followed by Boneferroni's comparison test, which revealed no significant differences. Source data are available online for this figure.

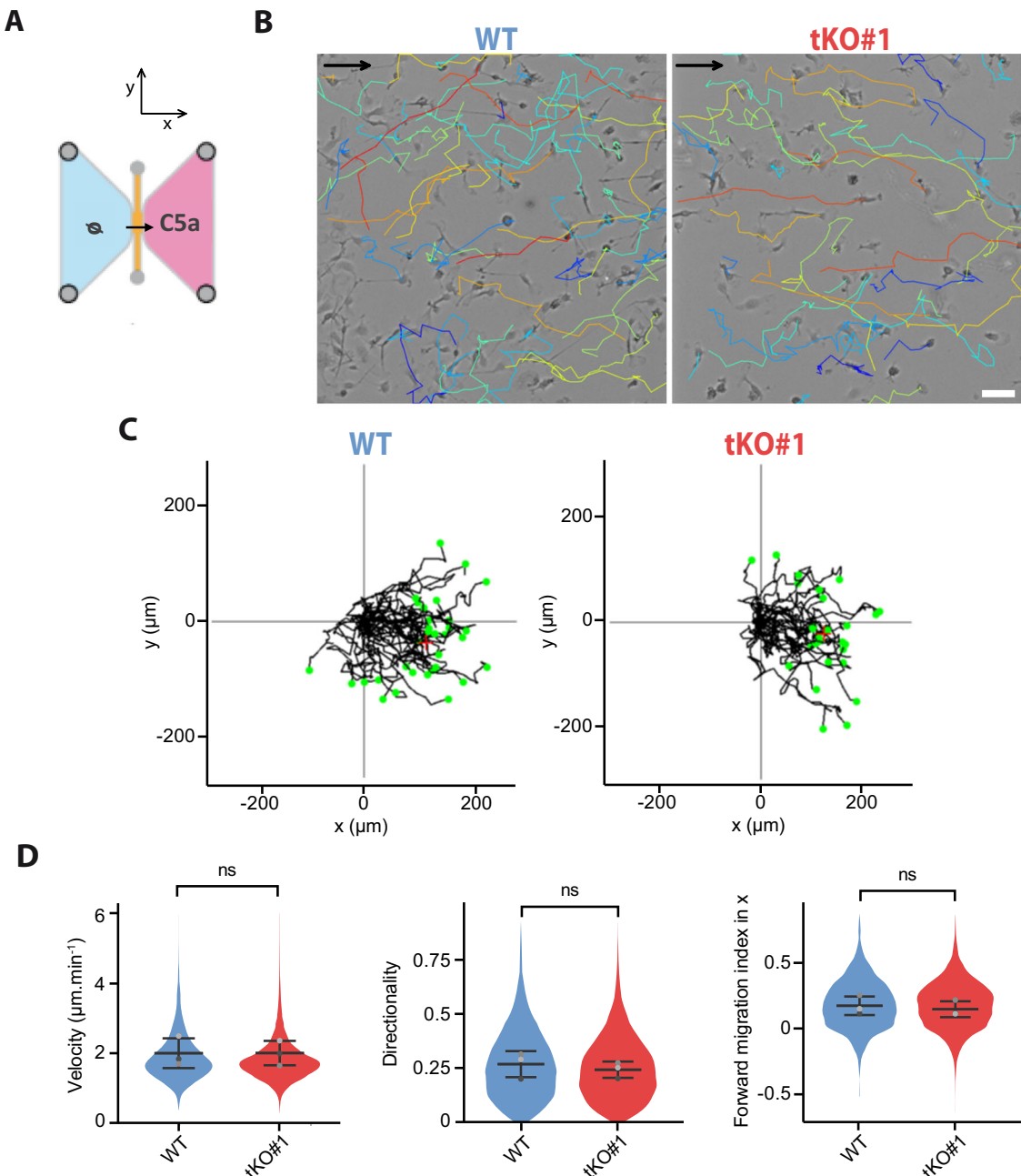

**Figure EV4. 2D chemotaxis of WT and ERM-tKO#1 macrophages toward a C5a gradient.**

(A) Schematic representation of 2D chemotaxis assay toward C5a. WT and ERM-tKO#1 cells migrating along a C5a gradient in the x axis. See also Movie EV9. (B) Snapshot of WT and ERM-tKO#1 macrophages migrating toward C5a gradient (on the right) with migratory tracks representing cell trajectories during 90 min. Tracks are color-coded according to their directionality. Scale bar: 50 μm. (C) Migratory tracks of WT and ERM-tKO#1 macrophages with origins set at (0,0). (D) Quantification of the median velocity, the directionality, and the forward migration index in the x axis (FMIx, used as a chemotaxis indicator) of each migratory track. The medians of 3 independent experiments are represented (gray points) and used for statistical analysis with a paired *t* test. Means and SD are shown.

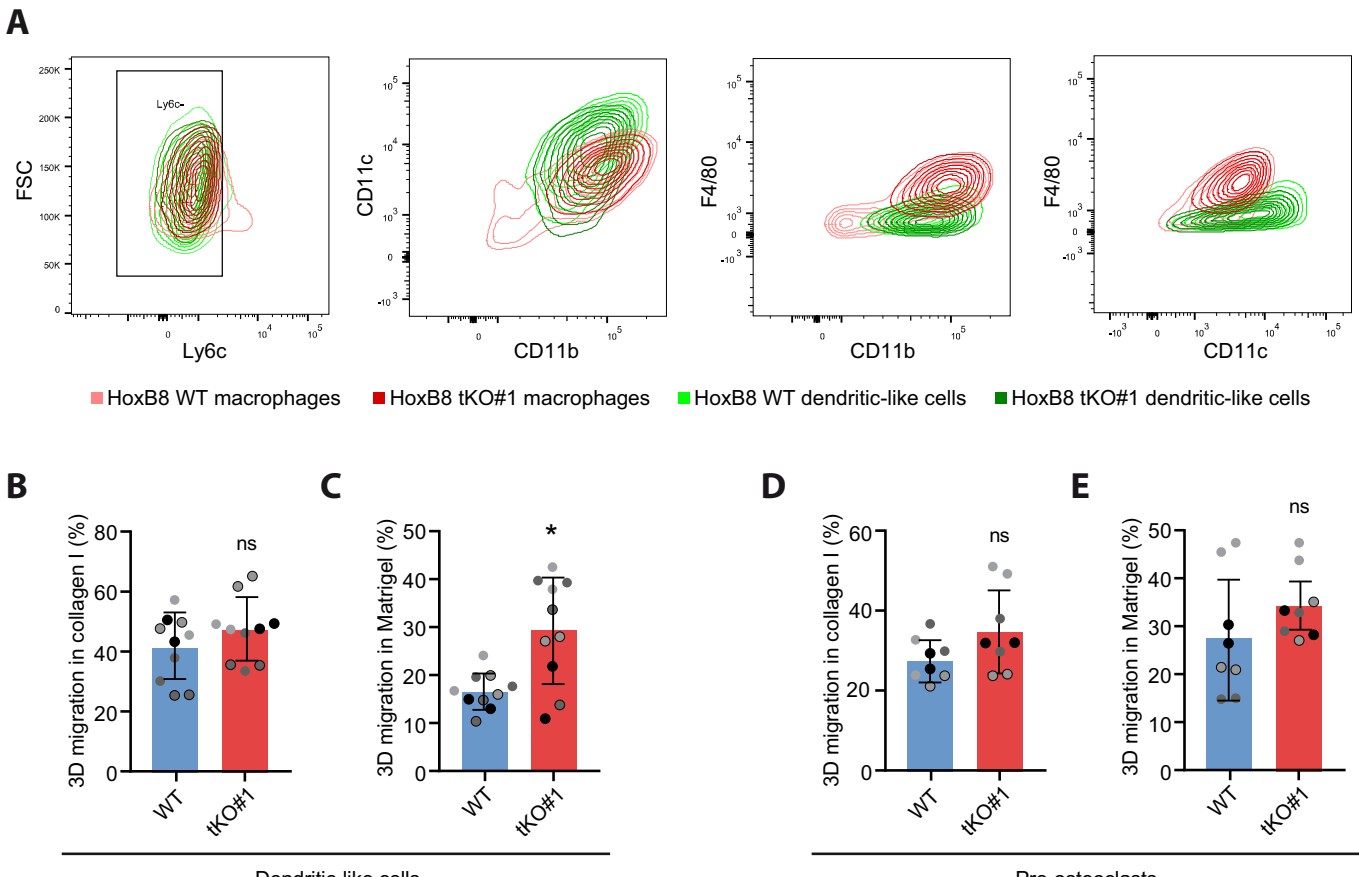

**Figure EV5. ERM inhibit the 3D mesenchymal migration of dendritic-like cells.**

(**A–C**) ERM-tKO affects the 3D migration through Matrigel of dendritic-like cells. (**A**) Differentiation of HoxB8 progenitors in dendritic-like cells. FACS analyses shows that HoxB8 progenitors differentiated with 40 ng/mL GM-CSF are Ly6C -, CD11b +, CD11c high, and F4/80 - dendritic-like cells (green), compared to M-CSF, which differentiates the same progenitors into Ly6C -, CD11b +, CD11c low and F4/80 + macrophages (red). (**B, C**) Percentages of migration of siRNA-treated HMDM inside collagen I (**B**) and Matrigel (**C**) are represented as follows: the technical replicates (dot) of 5 independent experiments (highlighted by different gray colors) are represented. The mean (bar) and SD from the 5 independent experiments are shown. Statistical analysis was done on the mean per experiment using a paired two-tailed *t* test. *$P < 0.05$. (**D, E**) ERM-tKO does not affect the 3D migration of pre-osteoclasts. Percentages of migration inside collagen I (**D**) and Matrigel (**E**) are represented as follows: the technical replicates (dot) of 4 independent experiments (highlighted by different gray colors) are represented. The mean (bar) and SD from the independent experiments are shown. Statistical analysis was done on the mean per experiment using a paired two-tailed *t* test.

