## [Peer Review File · The EMBO Journal]

Ezrin, radixin, and moesin are dispensable for macrophage migration and cellular cortex mechanics

Perrine Verdys, Javier Rey Barroso, Joseph Vermeil, Martin Bergert, Thibaut Sanchez, Arnaud Métais, Thomas Mangeat, Elisabeth Bellard, Claire Bigot, Jean-Philippe Girard, Isabelle Maridonneau-Parini, Christel Vérollet, Frederic Lagarrigue, Alba Diz-Munoz, Julien Heuvingh, Matthieu Piel, Olivia Du Roure, Veronique Le Cabec, Sébastien Carreno, Renaud Poincloux, Adeline Girel, Catherine Astarie-Dequeker, and Arnaud Labrousse

Corresponding authors: Renaud Poincloux (Renaud.Poincloux@ipbs.fr) , Sébastien Carreno (sebastien.carreno@umontreal.ca), Veronique Le Cabec (veronique.le-cabec@ipbs.fr)

Review Timeline:

Transferred from Review Commons:	24th Oct 23
Editorial Correspondence:	2nd Nov 23
Authors' Correspondence:	10th Nov 23
Editorial Decision:	10th Nov 23
Revision Received:	6th May 24
Editorial Decision:	27th May 24
Revision Received:	17th Jun 24
Accepted:	1st Jul 24

Editor: Ieva Gailite

Transaction Report:

Review
COMMONS

This manuscript was transferred to The EMBO Journal following peer review at Review Commons.

Review #1

1. Evidence, reproducibility and clarity:

Evidence, reproducibility and clarity (Required)

In the manuscript, the authors systematically test the role of ERM proteins in macrophages using RNA silencing as well as the genetic knockout approaches. Previous studies have highlighted the fundamental importance of ERM proteins as structural and regulatory components of the cell cortex governing several essential functions such as the generation of surface features such as filopodia, maintenance of cortex-plasmamembrane attachment, bleb retraction, cortical mechanics, and cell migration. The authors performed a series of experiments to comprehensively test each of these functions (including cell migration in 2D surface and 3D matrix *in vitro*, *ex vivo* on tumor implants, as well as *in vivo*) and found that none of these are significantly affected when ERM proteins are downregulated in macrophages. Overall, the paper is solid, the experiments are well-designed and conclusive, and the manuscript is written well.

I have no significant concerns with the study. My only experimental suggestion is related to a previously shown function of ERM protein in macrophages- the ERM proteins play an important role in phagosome maturation in macrophages (Defacque et al., EMBO, 2000; Lars-Peter et al., PNAS, 2006; Mylvaganam et al., Current Biology, 2021). It would be nice if authors could explore this phenotype in their perturbation system.

A minor concern with the study is, as the authors have already pointed out, that ERM proteins may still be required for some functions in macrophages under specific (environmental?) conditions. It is of course impossible to experimentally test all possible conditions that may involve ERMs, however, the authors should include a note on the hypothetical conditions that may require ERMs in macrophages. They should also discuss possible hypothetical reasons why macrophages may have evolved a cortex that does not rely on ERM proteins for specific functions. Overall, a more extended discussion on the role of ERM proteins (or the lack of them) in macrophages is required.

2. Significance:

Significance (Required)

The manuscript is important on many accounts: The ERM proteins are considered crucial membrane-cytoskeletal linkers in many cellular systems. The study presents a surprising finding that cortical phenomena requiring membrane-cytoskeletal attachment do not essentially need ERM proteins providing a fundamental conceptual advance. The results from this study will also inform both experimental as well as theoretical studies of cortical organization and dynamics in the future. Furthermore, overexpressed mutant forms of ERMs are used as sensors as well as perturbing agents of cortical actin dynamics in many cellular systems. These utilities can now be further substantiated and if required, revised in light of the results from this study.

I am an immune cell biologist specializing in early lymphocyte activation and cytoskeleton dynamics.

3. How much time do you estimate the authors will need to complete the suggested revisions:

Estimated time to Complete Revisions (Required)

(Decision Recommendation)

Between 1 and 3 months

Yes

Review #2

1. Evidence, reproducibility and clarity:

Evidence, reproducibility and clarity (Required)

****Summary****

ERM proteins are known to play a central role in linking the cortical actin cytoskeleton with the plasma membrane, which is involved in regulating a diverse range of actin-rich membrane structures. The authors question the role that ERM proteins play in regulating cell shape changes and migration, specifically in macrophages. To test this, they designed an approach to systematically delete each ERM in macrophages - followed by the production of a triple-ERM ko line (tKO) using HoxB8 myeloid progenitor cells. The tKO line was subjected to a series of in vitro and in vivo experiments - all of which involve a series of imaging techniques to monitor

membrane dynamics, protein subcellular organisation and cellular behaviours (e.g. rolling fraction, sticking fraction and chemotaxis). Their overall conclusion is that ERM are dispensable for macrophage membrane structures and migration.

****General comments.****

The experiments are very well executed. The manuscript in short demonstrates that the ERM proteins are dispensable for macrophage migration (both in 2D and 3D contexts), but there is very little beyond this work that points to what they might be doing instead. In this regard, given that the focus is exclusively on macrophage migration, the work comes across as quite specialised.

The biggest concern I have is with the *in vivo* part. It should be noted that the work outlined in the manuscript does not actually address diapedesis, which is monitoring transmigration from the blood into tissue. Rolling and sticking do not define diapedesis. The experiments that the authors have conducted may have captured diapedesis events, but that very much depends on the length of time that the IVM was conducted. The authors would need to qualify their claims in this regard. Removing this work altogether would not lessen the impact, given that diapedesis is not shown. The work would therefore be very much *in vitro/ex vivo*.

****Specific questions****

How sure are the authors that they are capturing these events in cremasteric venules? Is there any sign of cells being trapped in the microcirculation? The reason for injecting macrophages intravenously is not explained. Are these experiments modelling intravascular (patrolling) macrophages? Monocytes will typically differentiate into macrophages in tissue. The fact that the cells are able to "roll" and "stick" suggests that they have the complimentary cell adhesion molecules, although this is not addressed in the study.

2. Significance:

Significance (Required)

The strength of the manuscript is based on the robust *in vitro* experiments, however such experiments are difficult to address *in vivo* - mainly because of the issue that macrophages (unless patrolling macrophages) are not a useful model to investigate for *ivm* experiments.

This would be of great interest to the macrophage field, which is quite limited in scope.

An advancement in the field would be to learn what is taking over the role of ERM in macrophages. As such, this becomes a report with a series of experiments to confirm that ERM are not involved.

3. How much time do you estimate the authors will need to complete the suggested revisions:

Estimated time to Complete Revisions (Required)

(Decision Recommendation)

Cannot tell / Not applicable

No

Review #3

1. Evidence, reproducibility and clarity:

Evidence, reproducibility and clarity (Required)

Verdys and colleagues report an elegant study in which the authors describe that ERM proteins are dispensable for migrating monocyte-derived macrophages. The methods are adequate and the results support the conclusions.

****Major points:****

- Although the authors demonstrate, by multiple methods, the dispensability of ERM proteins in the migration of macrophages derived from monocytes, the role of these proteins must also be evaluated in the phagocytosis process (another relevant functional aspect of macrophages).
- How is the activation of key downstream targets of ERM proteins involved in macrophage migration in KO models?

2. Significance:

Significance (Required)

Advance: The present study fills a gap in the participation of ERM proteins in cell migration. The results obtained on the dispensability of these proteins in macrophage migration can pave avenues for identifying new processes and proteins associated with migration in this context.

Audience: The audience for this study is very broad.

My expertise: I have expertise in cellular and molecular biology with a focus on processes associated with cancer. Among the numerous research fronts of the group led by me, we recently identified the EZR gene (which encodes the ezrin protein) as a prognostic marker and molecular target in acute leukemias.

3. How much time do you estimate the authors will need to complete the suggested revisions:

Estimated time to Complete Revisions (Required)

(Decision Recommendation)

Between 3 and 6 months

Yes

General Statements

Ezrin, Radixin, and Moesin (ERMs) serve as crucial cytoskeletal linker proteins, connecting the actin cytoskeleton to the plasma membrane upon activation. ERMs are essential regulators of cell morphogenesis across every cell types reported so far, and have been implicated in vital cellular functions such as migration, and invasion. In our study, we discovered that ERMs are dispensable for the cortical organization of macrophages. In accordance with this surprising finding, we found that the migration of macrophages was not affected upon knock-out of the three ERMs. Our findings challenge the prevailing belief that ERMs universally regulate cortical organization. Instead, they indicate that the actin cortex of macrophages has evolved to possess a high degree of adaptability and plasticity, enabling these immune cells to function independently of ERM proteins.

We thank the editors of Review Commons that handled our manuscript and all three reviewers for their positive assessment of our manuscript and for their constructive suggestions.

Description of the planned revisions

Reviewer #1 (Evidence, reproducibility and clarity (Required)):

In the manuscript, the authors systematically test the role of ERM proteins in macrophages using RNA silencing as well as the genetic knockout approaches. Previous studies have highlighted the fundamental importance of ERM proteins as structural and regulatory components of the cell cortex governing several essential functions such as the generation of surface features such as filopodia, maintenance of cortex-plasma membrane attachment, bleb retraction, cortical mechanics, and cell migration. The authors performed a series of experiments to comprehensively test each of these functions (including cell migration in 2D surface and 3D matrix in vitro, ex vivo on tumor implants, as well as in vivo) and found that none of these are significantly affected when ERM proteins are downregulated in macrophages. Overall, the paper is solid, the experiments are well-designed and conclusive, and the manuscript is written well.

We thank the reviewer for these encouraging comments.

I have no significant concerns with the study. My only experimental suggestion is related to a previously shown function of ERM protein in macrophages- the ERM proteins play an important role in phagosome maturation in macrophages (Defacque et al., EMBO, 2000; Lars-Peter et al., PNAS, 2006; Mylvaganam et al., Current Biology, 2021). It would be nice if authors could explore this phenotype in their perturbation system.

We thank the reviewer for this valuable suggestion. ERM proteins have indeed been proposed as important for macrophage phagocytosis. Importantly, their necessity for the early steps of this process is still debated, as conclusions differ depending on the cellular model used and the type of particle to be internalised (Erwig et al., PNAS USA 2006; Di Pietro et al., Sci. Rep. 2017; Gomez and Descoteaux, Biochem. Biophys. Res. Commun. 2018; Mu et al., Nat. Commun. 2018; Okazaki et al., J. Physiol. Sci. 2020). While the implication of ERMs in the early steps of phagocytosis remains controversial, there seems to be a consensus to implicate ezrin and moesin in phagosome maturation (Defacque et al. EMBO 2000 ; Erwig et al., PNAS USA 2006; Marion et al., Traffic 2011; Gomez and Descoteaux, Biochem. Biophys. Res. Commun. 2018).

We have already started addressing the ability of ERM-depleted macrophages to perform phagocytosis. In particular, we quantified the dynamics of phagocytosis of ovalbumin-coated or IgG-

opsonized polystyrene beads, which did not reveal any difference between WT and ERM-depleted macrophages (Fig.1).

Figure 1: Phagocytosis by WT and ERM-tKO macrophages. *HoxB8* macrophages cultured on glass coverslips were exposed to IgG-beads (MOI 4:1) or OVA-beads (MOI 24:1). The beads were centrifuged to synchronize phagocytosis and the experiments were carried out for the indicated times (5 to 120 min).

(A) Representative images of fluorescence microscopy of macrophages exposed to OVA-beads for 60 min are shown. Beads that remained outside the cells were distinguished from ingested beads using anti-ovalbumin antibodies and TRITC-coupled secondary antibodies. Beads inside cells are magenta, beads outside cells are green, F-actin is shown in white and nuclei in blue. Scale bars: 50 μm or 10 μm for zooms.

(B-E) The percentages of phagocytosing cells (B, D) and percentages of fully internalized beads (C, E) were quantified. Results are expressed as mean \pm SD of at least 2600 cells/time point from 4 independent experiments; and analyzed with two-way Anova followed by Bonferroni's comparison test.

Proposed revision: We propose to include in the manuscript our quantification of IgG-coated and non-coated phagocytosis, and evaluate whether phago-lysosome fusion is delayed in ERM-depleted macrophages.

A minor concern with the study is, as the authors have already pointed out, that ERM proteins may still be required for some functions in macrophages under specific (environmental?) conditions. It is of course impossible to experimentally test all possible conditions that may involve ERMs, however, the authors should include a note on the hypothetical conditions that may require ERMs in macrophages. They should also discuss possible hypothetical reasons why macrophages may have evolved a cortex that does not rely on ERM proteins for specific functions. Overall, a more extended discussion on the role of ERM proteins (or the lack of them) in macrophages is required.

As suggested, in the revised version of the manuscript we will add a more extensive discussion of the role of ERM proteins in macrophages, and in particular the hypothetical conditions that might require their presence, as well as the reasons why macrophages have developed a particular cortex.

Reviewer #1 (Significance (Required)):

The manuscript is important on many accounts: The ERM proteins are considered crucial membrane-cytoskeletal linkers in many cellular systems. The study presents a surprising finding that cortical phenomena requiring membrane-cytoskeletal attachment do not essentially need ERM proteins providing a fundamental conceptual advance. The results from this study will also inform both experimental as well as theoretical studies of cortical organization and dynamics in the future. Furthermore, overexpressed mutant forms of ERMs are used as sensors as well as perturbing agents of cortical actin dynamics in many cellular systems. These utilities can now be further substantiated and if required, revised in light of the results from this study.

I am an immune cell biologist specializing in early lymphocyte activation and cytoskeleton dynamics.

We would like to thank the reviewer for pointing out the importance of our work for our understanding of the function of the cellular cortex, and for highlighting the fact that it may lead to a reinterpretation of the results obtained using ERM mutants.

Reviewer #2 (Evidence, reproducibility and clarity (Required)):

Summary

ERM proteins are known to play a central role in linking the cortical actin cytoskeleton with the plasma membrane, which is involved in regulating a diverse range of actin-rich membrane structures. The

authors question the role that ERM proteins play in regulating cell shape changes and migration, specifically in macrophages. To test this, they designed an approach to systematically delete each ERM in macrophages - followed by the production of a triple-ERM ko line (tKO) using HoxB8 myeloid progenitor cells. The tKO line was subjected to a series of in vitro and in vivo experiments - all of which involve a series of imaging techniques to monitor membrane dynamics, protein subcellular organisation and cellular behaviours (e.g. rolling fraction, sticking fraction and chemotaxis). Their overall conclusion is that ERM are dispensable for macrophage membrane structures and migration.

General comments.

The experiments are very well executed. The manuscript in short demonstrates that the ERM proteins are dispensable for macrophage migration (both in 2D and 3D contexts), but there is very little beyond this work that points to what they might be doing instead. In this regard, given that the focus is exclusively on macrophage migration, the work comes across as quite specialised.

We thank the reviewer for appreciating the quality of our work.

We respectfully disagree to their assessment of the limited scope of our findings. Given the crucial importance of the migration of macrophages for so many of our body's functions, our findings will have a wide-ranging impact. Furthermore, and as acknowledged by Reviewers 1 and 3, we believe that the discovery that ERMs do not play a universal role in cortical mechanics and in cell migration, as hitherto believed, reaches a much wider scientific audience than that of the macrophage field. By proposing a unique research model (a triple KO for ERMs), our work allows to question many studies carried out with less direct molecular tools, such as the use of drugs or mutants of ERMs.

We acknowledge the fact that although our data convincingly demonstrate that the ERM proteins are dispensable for macrophage migration, they do not reveal alternative functions for these proteins. We agree that it could be interesting to search for alternative functions for ERM proteins in macrophages in future studies. However, we believe that such studies are out of the scope of the present manuscript.

The biggest concern I have is with the in vivo part. It should be noted that the work outlined in the manuscript does not actually address diapedesis, which is monitoring transmigration from the blood into tissue. Rolling and sticking do not define diapedesis. The experiments that the authors have conducted may have captured diapedesis events, but that very much depends on the length of time that the IVM was conducted. The authors would need to qualify their claims in this regard. Removing this work altogether would not lessen the impact, given that diapedesis is not shown. The work would therefore be very much in vitro/ex vivo.

We agree with the reviewer that, due to technical limitations, we only measured the rolling and sticking capacity of the +/-ERM cells and did not measure diapedesis directly. Following Reviewer's comments, we have thus modified the text of the manuscript and no longer use the term 'diapedesis' to describe our in vivo intravital imaging studies.

We also clarified the fact that we did not inject differentiated macrophages into the circulation, but macrophage precursors obtained by the treatment of progenitors with a 1 day treatment only (and not a 7 day treatment) with 20 ng/mL M-CSF.

Here, highlighted in yellow, are the changes to the text (in the Introduction, Results, Methods and Legends sections):

Introduction, p3:

“Surprisingly, we found that ERMs are dispensable for macrophages to migrate in diverse contexts, including in vitro 2D migration and 3D invasion of extracellular matrix, ex vivo tissue infiltration through healthy dermis and tumor tissue, and for the in vivo adhesion of macrophage precursors to an activated endothelium.”

Results, p6:

“ERM tKo cells without ezrin, radixin, and moesin exhibit no impairment in their ability to adhere to vascular endothelium in vivo and infiltrate the ear derma or fibrosarcoma.

To further investigate the migratory properties of ERM-deficient cells in vivo, we first assessed their ability to adhere to activated vascular endothelium into mice bearing a fibrosarcoma (Gui et al., 2018).”

Results, p8:

“Our study uncovered a surprising finding: ezrin, radixin and moesin are dispensable for key aspects of macrophage behavior, including the formation of lamellipodia and filopodia, the dynamics of membrane ruffles and podosomes, migration in vitro (in 2D or 3D matrices) and ex vivo (into dermis or tumor tissues) as well as for the in vivo adhesion of macrophage precursors to activated vascular endothelium.”

Methods, p14:

“In vivo analysis of adhesion to vascular endothelium with wide-field intravital microscopy”

And

“HoxB8-progenitors were directed towards monocyte/macrophage differentiation using a 1 day treatment with 20 ng/mL mouse M-CSF.”

Figure 4 legend, p33:

“Fig. 4: ERM tKo cells have no defect in adhesion to vascular endothelium in vivo and infiltrate tissues explants ex vivo

A. In vivo adhesion to vascular endothelium

Fibrosarcoma cells were injected into the flank of a mice. After a week, tumor was exposed for intravital microscopy, and the femoral artery of recipient mice was catheterized for injection of exogenous cells. Differentially labeled WT and TKO-ERM macrophage precursors were injected in the blood and their behaviour in tumor blood vessels was assessed by real-time imaging. Rolling fractions were quantified as the percentage of rolling cells in the total flux of cells in each blood vessel, and sticking fractions were quantified as the percentage of rolling cells that firmly adhered for a minimum of 30 seconds.”

Proposed revision: We propose to keep the results of the *in vivo* experiments in the manuscript, including the modifications proposed by the reviewer and listed above.

Specific questions

How sure are the authors that they are capturing these events in cremasteric venules?

As described in the Results and Methods section, these measurements were not captured in cremasteric venules but in fibrosarcoma tumour blood vessels, where we have previously demonstrated strong recruitment of circulating monocytes to infiltrate tumor tissue (Gui et al., Cancer Immunol. Res. 2018).

Is there any sign of cells being trapped in the microcirculation?

The diameter of the tumor blood vessels analysed is consistent with tumor post-capillary venules, and we have not seen cells trapped in these tumor blood vessels.

The reason for injecting macrophages intravenously is not explained.

We injected cells intravenously in order to compare their capacity to adhere to activated tumor blood vessels by intravital microscopy. This is now clarified in the corresponding result section (p6):

“For that purpose, one day differentiated wild-type or ERM-deficient cells were fluorescently labelled with two different cell trackers, mixed in a 1:1 ratio, and co-injected intra-arterially into recipient mice in order to analyse their behaviour in tumor blood vessels by intravital microscopy.”

Are these experiments modelling intravascular (patrolling) macrophages? Monocytes will typically differentiate into macrophages in tissue.

We again apologize for the lack of clarity. In these experiments, we did not inject fully differentiated (seven days) macrophages but progenitors directed towards monocyte/macrophage differentiation using a 1 day treatment with 20 ng/mL mouse M-CSF. We believe that these experiments model the adhesion/recruitment of monocytes by activated vascular endothelium in the tumor microenvironment.

The fact that the cells are able to "roll" and "stick" suggests that they have the complimentary cell adhesion molecules, although this is not addressed in the study.

We agree with the reviewer. Our intravital microscopy analyses indicate that the injected cells have the complementary cell adhesion molecules for firm adhesion to activated tumor blood vessels. Importantly, our data clearly demonstrate that the capacity of ERM-tKO cells to bind vascular endothelium in the tumor microenvironment is similar to that of WT cells (Fig. 4A).

Reviewer #2 (Significance (Required)):

The strength of the manuscript is based on the robust in vitro experiments, however such experiments are difficult to address in vivo - mainly because of the issue that macrophages (unless patrolling macrophages) are not a useful model to investigate for ivm experiments.

We thank the reviewer for recognizing the robustness of our *in vitro* experiments. We fully agree with the reviewer that the *in vivo* experiments are more challenging and that the behaviour of monocytes/(patrolling) macrophages is difficult to mimic *in vivo*. However, we believe that our intravital microscopy analyses are important because they demonstrate that ERM-tKO cells retain the capacity to bind firmly (sticking) to activated tumor blood vessels *in vivo*.

This would be of great interest to the macrophage field, which is quite limited in scope. An advancement in the field would be to learn what is taking over the role of ERM in macrophages. As such, this becomes a report with a series of experiments to confirm that ERMs are not involved.

Again, we respectfully disagree with the reviewer, as this work goes against the dogma that ERMs are generally the most important mechanical links between the plasma membrane and the cytoskeleton.

By clearly establishing that this is not the case in macrophages, cells whose importance for our immunity justifies the importance of their investigation, this study could make it possible to reconsider the functioning of the cellular cortex and the role of ERMs in other cellular systems.

Reviewer #3 (Evidence, reproducibility and clarity (Required)):

Verdys and colleagues report an elegant study in which the authors describe that ERM proteins are dispensable for migrating monocyte-derived macrophages. The methods are adequate and the results support the conclusions.

We thank the reviewer for these very supportive comments.

Major points:

- Although the authors demonstrate, by multiple methods, the dispensability of ERM proteins in the migration of macrophages derived from monocytes, the role of these proteins must also be evaluated in the phagocytosis process (another relevant functional aspect of macrophages).

This is an excellent suggestion, which should make it possible to clarify the role of ERMs in this important function of macrophages.

ERM proteins have indeed been proposed as important for macrophage phagocytosis. Importantly, their necessity for the early steps of this process is still debated, as conclusions differ depending on the cellular model used and the type of particle to be internalised (Erwig et al., PNAS USA 2006; Di Pietro et al., Sci. Rep. 2017; Gomez and Descoteaux, Biochem. Biophys. Res. Commun. 2018; Mu et al., Nat. Commun. 2018; Okazaki et al., J. Physiol. Sci. 2020). While the implication of ERMs in the early steps of phagocytosis remains controversial, there seems to be a consensus to implicate ezrin and moesin in phagosome maturation (Defacque et al. EMBO 2000 ; Erwig et al., PNAS USA 2006; Marion et al., Traffic 2011; Gomez and Descoteaux, Biochem. Biophys. Res. Commun. 2018).

We have already started addressing the ability of ERM-depleted macrophages to perform phagocytosis. In particular, we quantified the dynamics of phagocytosis of ovalbumin-coated or IgG-opsonized polystyrene beads, which did not reveal any difference between WT and ERM-depleted macrophages (Fig.1).

Figure 1: Phagocytosis by WT and ERM-tKO macrophages. *HoxB8* macrophages cultured on glass coverslips were exposed to IgG-beads (MOI 4:1) or OVA-beads (MOI 24:1). The beads were centrifuged to synchronize phagocytosis and the experiments were carried out for the indicated times (5 to 120 min).

(A) Representative images of fluorescence microscopy of macrophages exposed to OVA-beads for 60 min are shown. Beads that remained outside the cells were distinguished from ingested beads using anti-ovalbumin antibodies and TRITC-coupled secondary antibodies. Beads inside cells are magenta, beads outside cells are green, F-actin is shown in white and nuclei in blue. Scale bars: 50 μm or 10 μm for zooms.

(B-E) The percentages of phagocytosing cells (B, D) and percentages of fully internalized beads (C, E) were quantified. Results are expressed as mean \pm SD of at least 2600 cells/time point from 4 independent experiments; and analyzed with two-way Anova followed by Bonferroni's comparison test.

Proposed revision: We propose to include in the manuscript our quantification of IgG-coated and non-coated phagocytosis, and evaluate whether phago-lysosome fusion is delayed in ERM-depleted macrophages.

- How is the activation of key downstream targets of ERM proteins involved in macrophage migration in KO models?

This is a very pertinent question. However, while ERMs have been described as being downstream of several signalling pathways, their own downstream targets are unfortunately poorly documented and, to our knowledge, none are known in macrophages.

In different cellular contexts, it has been proposed that ERMs regulate PI3K (Gautreau et al. PNAS USA 1999), Ras (Sperka et al. Plos One 2011) or that they are involved in the initiation of protein translation (Briggs et al. Neoplasia 2012), but these results have not yet been confirmed and we believe they are outside the scope of this study.

During macrophage migration, we consider that their obvious main target is cortical actin, and demonstrate in this manuscript that the functional coupling between actin and the plasma membrane is not affected by full ERM knockout.

Reviewer #3 (Significance (Required)):

Advance: The present study fills a gap in the participation of ERM proteins in cell migration. The results obtained on the dispensability of these proteins in macrophage migration can pave avenues for identifying new processes and proteins associated with migration in this context.

Audience: The audience for this study is very broad.

We again thank the reviewer for recognising the importance of this work for the understanding of cell migration.

My expertise: I have expertise in cellular and molecular biology with a focus on processes associated with cancer. Among the numerous research fronts of the group led by me, we recently identified the EZR gene (which encodes the ezrin protein) as a prognostic marker and molecular target in acute leukemias.

Description of the revisions that have already been incorporated in the transferred manuscript

In the revised version of the article, we have taken into account all relevant changes proposed by the reviewers. We modified the text of the manuscript and no longer use the term 'diapedesis' to describe our in vivo intravital imaging studies, and clarified the fact that we did not inject differentiated macrophages into the circulation, but macrophage precursors obtained by the treatment of progenitors with a 1 day treatment only (and not a 7 day treatment) with 20 ng/mL M-CSF.

Here, highlighted in yellow, are the changes to the text (in the Introduction, Results, Methods and Legends sections):

Introduction, p3:

“Surprisingly, we found that ERMs are dispensable for macrophages to migrate in diverse contexts, including in vitro 2D migration and 3D invasion of extracellular matrix, ex vivo tissue infiltration through healthy dermis and tumor tissue, and for the in vivo **adhesion of macrophage precursors to an activated endothelium.**”

Results, p6:

“ERM tKo cells without ezrin, radixin, and moesin exhibit no impairment in their ability to adhere to vascular endothelium in vivo and infiltrate the ear derma or fibrosarcoma.

To further investigate the migratory properties of ERM-deficient cells in vivo, we first assessed their ability **to adhere to activated vascular endothelium** into mice bearing a fibrosarcoma (Gui et al., 2018). For that purpose, one day differentiated wild-type or ERM-deficient cells were fluorescently labelled with two different cell trackers, mixed in a 1:1 ratio, and co-injected intra-arterially into recipient mice **in order to analyse their behaviour in tumor blood vessels by intravital microscopy.**”

Results, p8:

“Our study uncovered a surprising finding: ezrin, radixin and moesin are dispensable for key aspects of macrophage behavior, including the formation of lamellipodia and filopodia, the dynamics of membrane ruffles and podosomes, migration in vitro (in 2D or 3D matrices) and ex vivo (into dermis or tumor tissues) as well as for the in vivo **adhesion of macrophage precursors to activated vascular endothelium).**”

Methods, p14:

“In vivo analysis of **adhesion to vascular endothelium** with wide-field intravital microscopy”

And

“HoxB8-progenitors were **directed towards monocyte/macrophage differentiation using a 1 day treatment** with 20 ng/mL mouse M-CSF.”

Figure 4 legend, p33:

“Fig. 4: ERM tKO cells have no defect in **adhesion to vascular endothelium in vivo and infiltrate tissues explants ex vivo**

A. In vivo adhesion to vascular endothelium

Fibrosarcoma cells were injected into the flank of a mice. After a week, tumor was exposed for intravital microscopy, and the femoral artery of recipient mice was catheterized for injection of exogenous cells. Differentially labeled WT and TKO-ERM macrophage precursors were injected in the blood and their behaviour in tumor blood vessels was assessed by real-time imaging. Rolling fractions were quantified as the percentage of rolling cells in the total flux of cells in each blood vessel, and sticking fractions were quantified as the percentage of rolling cells that firmly adhered for a minimum of 30 seconds.”

Dear Dr. Poincloux,

Thank you for submitting your Review Commons manuscript to The EMBO Journal. I have now read your manuscript, the reviewer comments and your revision proposal, as well as discussed your manuscript with external scientific advisors.

I appreciate that your study shows that the triple knockout of ERM family proteins ezrin, radixin and moesin in macrophages does not affect their actin-based protrusion formation, cell morphology, actin cortex mechanics, and migration in various environments, including in the mouse endothelium. I appreciate the unexpected nature of this finding, as also echoed in the comments by reviewers #1 and #3. However, reviewer #2 also indicates that further insights into the specificity of the findings to this particular cell type would be needed, and also states that the alternative mechanisms replacing ERM protein function in macrophages remain unclear.

Due to the rather brief nature of the reports and the concerns indicated by reviewer #2, I have consulted with two scientific advisors familiar with immune cell migration and actin cortex function, as well as our journal. Both experts found that further insights would be needed for consideration here along some of the avenues outlined below:

- Assessment of the membrane-to-cortex attachment properties in macrophages vs other cell types - are there differences that would explain the lacking role of ERM proteins in this cell type?
- Does manipulation of membrane-cortex attachment, e.g., with the artificial linker generated in the Diz Munoz group, alter macrophage migratory properties per se?
- Is ERM protein independence also extendable to other immune cells, e.g., dendritic cells? Differentiation of the triple KO Hoxb8 cells into dendritic cells could allow testing their migratory behaviour.
- Does inducible triple ERM protein knockout or depletion to reduce compensation effects exhibit a similar lack of phenotype?

I would appreciate if you could let me know whether you would be able to expand the study in one or several of these directions during a revision. A brief response to would be very helpful for the final editorial decision.

Please feel free to contact me if you have any questions regarding this pre-decision consultation approach. I look forward to your response.

With best regards,

Ieva

Ieva Gailite, PhD
Senior Scientific Editor
The EMBO Journal
Meyerohofstrasse 1
D-69117 Heidelberg
Tel: +4962218891309
i.gailite@embojournal.org

Dear Ieva,

Sorry for the delay in replying, but I was waiting for some information from my collaborators.

We are very pleased that you are interested in our work.

All four of your advisors' proposals are relevant, and we want to address them:

- Assessment of the membrane-to-cortex attachment properties in macrophages vs other cell types - are there differences that would explain the lacking role of ERM proteins in this cell type?

*Our collaborator Alba Diz Munoz has already measured membrane-cortex attachment in different cell models, and we could discuss these different measurements in this article.

- Does manipulation of membrane-cortex attachment, e.g., with the artificial linker generated in the Diz Munoz group, alter macrophage migratory properties per se?*

We could have macrophages express this linker and measure the effect of this expression on the 3D migration of macrophages.

- Is ERM protein independence also extendable to other immune cells, e.g., dendritic cells? Differentiation of the triple KO Hoxb8 cells into dendritic cells could allow testing their migratory behaviour.*

We could differentiate our control and triple KO progenitors in DC and test their 3D migration.

- Does inducible triple ERM protein knockout or depletion to reduce compensation effects exhibit a similar lack of phenotype?*

This point is more technically challenging and is likely to require more time. We would try to limit compensation, either by a triple siRNA or via an inducible KO, and we would also test the 3D migration of macrophages.

Please keep us informed of your decision,

With best regards,
Renaud

Dear Renaud,

I am glad to hear that you are willing to perform additional experiments to tackle the points raised by reviewer #2 and our external advisors. Based on your experimental outline provided in the revision plan and during the pre-decision consultation, I would like to invite you to submit a revised manuscript.

We generally allow three months as standard revision time. As a matter of policy, competing manuscripts published during this period will not negatively impact on our assessment of the conceptual advance presented by your study. However, please contact me as soon as possible upon publication of any related work to discuss the appropriate course of action. Should you foresee a problem in meeting this three-month deadline, please let us know in advance in order to arrange an extension.

When preparing your letter of response to the referees' comments, please bear in mind that this will form part of the Review Process File and will therefore be available online to the community. For more details on our Transparent Editorial Process, please visit our website: <https://www.embopress.org/page/journal/14602075/authorguide#transparentprocess>. Please also see the attached instructions for further guidelines on preparation of the revised manuscript.

Please feel free to contact me if have any further questions regarding the revision. Thank you for the opportunity to consider your work for publication. I look forward to receiving your revision.

With best regards,

leva

leva Gailite, PhD
Senior Scientific Editor
The EMBO Journal
Meyerhofstrasse 1
D-69117 Heidelberg
Tel: +4962218891309
i.gailite@embojournal.org

Please remember: Digital image enhancement is acceptable practice, as long as it accurately represents the original data and conforms to community standards. If a figure has been subjected to significant electronic manipulation, this must be noted in the

figure legend or in the 'Materials and Methods' section. The editors reserve the right to request original versions of figures and the original images that were used to assemble the figure.

We realize that it is difficult to revise to a specific deadline. In the interest of protecting the conceptual advance provided by the work, we recommend a revision within 3 months (8th Feb 2024). Please discuss the revision progress ahead of this time with the editor if you require more time to complete the revisions. Use the link below to submit your revision:

Link Not Available

Rev_Com_number: RC-2023-02102

New_manu_number: EMBOJ-2023-115975

Corr_author: Poincloux

Title: The membrane-actin linkers ezrin, radixin, and moesin are dispensable for macrophage migration and cortex mechanics.

General Statements

Ezrin, Radixin, and Moesin (ERMs) serve as crucial cytoskeletal linker proteins, connecting the actin cytoskeleton to the plasma membrane upon activation. ERMs are essential regulators of cell morphogenesis across every cell types reported so far, and have been implicated in vital cellular functions such as migration, and invasion. In our study, we discovered that ERMs are dispensable for the cortical organization of macrophages. In accordance with this surprising finding, we found that the migration of macrophages was not affected upon knock-out of the three ERMs. Our findings challenge the prevailing belief that ERMs universally regulate cortical organization. Instead, they indicate that the actin cortex of macrophages has evolved to possess a high degree of adaptability and plasticity, enabling these immune cells to function independently of ERM proteins.

We thank all three reviewers and EMBOJ scientific advisors for their positive assessment of our manuscript and for their constructive suggestions.

We have renamed the additional figures into Expanded View (EV) or Appendix Figures and provided the source data as requested by EMBOJ editorial policy. We were helped for the requested experiments by Adeline Girel, Catherine Astarie-Dequeker and Arnaud Labrousse, and they are now listed as co-authors of this publication.

Description of the planned revisions

Reviewer #1 (Evidence, reproducibility and clarity (Required)):

In the manuscript, the authors systematically test the role of ERM proteins in macrophages using RNA silencing as well as the genetic knockout approaches. Previous studies have highlighted the fundamental importance of ERM proteins as structural and regulatory components of the cell cortex governing several essential functions such as the generation of surface features such as filopodia, maintenance of cortex-plasma membrane attachment, bleb retraction, cortical mechanics, and cell migration. The authors performed a series of experiments to comprehensively test each of these functions (including cell migration in 2D surface and 3D matrix in vitro, ex vivo on tumor implants, as well as in vivo) and found that none of these are significantly affected when ERM proteins are downregulated in macrophages. Overall, the paper is solid, the experiments are well-designed and conclusive, and the manuscript is written well.

We thank the reviewer for these encouraging comments.

I have no significant concerns with the study. My only experimental suggestion is related to a previously shown function of ERM protein in macrophages- the ERM proteins play an important role in phagosome maturation in macrophages (Defacque et al., EMBO, 2000; Lars-Peter et al., PNAS, 2006; Mylvaganam et al., Current Biology, 2021). It would be nice if authors could explore this phenotype in their perturbation system.

We thank the reviewer for this valuable suggestion. ERM proteins have indeed been proposed as important for macrophage phagocytosis. Importantly, their necessity for the early steps of this process is still debated, as conclusions differ depending on the cellular model used and the type of particle to be internalised (Erwig et al., PNAS USA 2006; Di Pietro et al., Sci. Rep. 2017; Gomez and Descoteaux, Biochem. Biophys. Res. Commun. 2018; Mu et al., Nat. Commun. 2018; Okazaki et al., J. Physiol. Sci. 2020). While the implication of ERMs in the early steps of phagocytosis remains controversial, there seems to be a consensus to implicate ezrin and moesin in phagosome maturation (Defacque et al. EMBO 2000 ; Erwig et al., PNAS USA 2006; Marion et al., Traffic 2011; Gomez and Descoteaux, Biochem. Biophys. Res. Commun. 2018).

We addressed the ability of ERM-depleted macrophages to perform phagocytosis. In particular, we quantified the dynamics of phagocytosis of ovalbumin-coated or IgG-opsonized polystyrene beads, which did not reveal any difference between WT and ERM-depleted macrophages (Fig.1).

Figure 1: Phagocytosis by WT and ERM-tKO macrophages. *HoxB8* macrophages cultured on glass coverslips were exposed to IgG-beads (MOI 4:1) or OVA-beads (MOI 24:1). The beads were centrifuged to synchronize phagocytosis and the experiments were carried out for the indicated times (5 to 120 min).

(A) Representative images of fluorescence microscopy of macrophages exposed to OVA-beads for 60 min are shown. Beads that remained outside the cells were distinguished from ingested beads using anti-ovalbumin antibodies and TRITC-coupled secondary antibodies. Beads inside cells are magenta, beads outside cells are green, F-actin is shown in white and nuclei in blue. Scale bars: 50 μ m or 10 μ m for zooms.

(B-E) The percentages of phagocytosing cells (**B, D**) and percentages of fully internalized beads (**C, E**) were quantified. Results are expressed as mean \pm SD of at least 2600 cells/time point from 4 independent experiments; and analyzed with two-way Anova followed by Bonferroni's comparison test.

These results are now included in the revised manuscript as Figure EV3.

We then addressed whether phago-lysosome fusion is delayed in ERM-depleted macrophages. In our hands, labelling signs of fusion of lysosomes with the phagosomes of IgG beads did not turn out to be particularly easy. We were able to demonstrate low accumulations at phagosomes, compared with signals at lysosomes, of LAMP1 or of LysoTracker (Fig.2), and our first analyses seem to confirm a slight inhibition of phago-lysosome fusion in the absence of ERMs (Fig.2), but these results remain to be confirmed and detailed and we believe that these results are too preliminary to be included in the revised manuscript.

Figure 2: Phago-lysosome fusion in WT and ERM-tKO macrophages. HoxB8 macrophages cultured on glass coverslips were exposed to IgG-beads (MOI 4:1) during 90 min.

(A) Fluorescence microscopy of macrophages having phagocytosed IgG beads (green) and stained for LAMP1 (magenta). The arrow points to a weak accumulation of LAMP1 staining around the phagocytosed bead.

(B) Fluorescence microscopy of macrophages having phagocytosed IgG beads (not shown) and stained with LysoTracker showing accumulation of LysoTracker around phagocytosed beads.

(C) The intensity of LysoTracker staining around phagocytosed particles were quantified in WT and tKO macrophages. Results are expressed as box and whiskers from 10 to 90 percentile from at least 300 cells per condition from 3 independent experiments. Medians from each experiment are also represented. Statistical analysis was done on the mean per experiments using a paired two-tailed t-test ($p=0,09$).

Scale bars: 50 μm (A; B), 5 μm (A'; B').

A minor concern with the study is, as the authors have already pointed out, that ERM proteins may still be required for some functions in macrophages under specific (environmental?) conditions. It is of course impossible to experimentally test all possible conditions that may involve ERMs, however, the authors should include a note on the hypothetical conditions that may require ERMs in macrophages. They should also discuss possible hypothetical reasons why macrophages may have evolved a cortex that does not rely on ERM proteins for specific functions. Overall, a more extended discussion on the role of ERM proteins (or the lack of them) in macrophages is required.

As suggested, we have added an extended discussion on the potential role of ERM proteins in macrophages in the revised version of the manuscript:

“Although our findings indicate that ERMs are not necessary for the standard functions of macrophages, we cannot completely discount that ERMs are necessary under specific conditions. Supporting this hypothesis, recent studies reported that a triple ERM knockout in RAW macrophages did not result in global morphological changes. Phagocytosis remained unchanged between control and triple ERM knockout RAW macrophages. However, an increase in CD44 mobility and an inhibition of podosome disappearance concomitant with IgG phagocytosis were reported in Raw ERM-KO macrophages (Ferling et al., 2024; Le et al., 2024). This suggests that ERMs, although not important for macrophage migration and phagocytosis may play a role in coordinating various macrophage functions.

Despite observing no differences in the behavior of ERM-tKO vs control macrophages across a wide array of assays, the absence of phenotypes upon ERM knockouts could be due to compensation by other proteins during the establishment of ERM-tKO HoxB8 cells. However, reducing the time between depletion of ERMs, either by siRNA or by CRISPR-Cas9, still did not yield any observable effects. This suggests that the absence of ERMs is unlikely to be compensated for by a gene regulation mechanism.

In any case, this study challenges the widely held belief that ERMs are universal regulators of cortical organization and cell migration (Akisawa et al., 1999; Arpin et al., 2011; Crepaldi et al., 1997; Louvet-Vallée, 2000; Valderrama et al., 2012). Instead, these findings suggest that the macrophage actin cortex is highly adaptable and plastic, enabling these immune cells to function independently of ERM proteins. Macrophages demonstrate unique plasticity among leukocytes, especially in their capacity to adapt their migration mode to the microenvironment (Van Goethem et al., 2010; Cougoule et al., 2012). Our results suggest that, in contrast to macrophage migration, DC migration is affected by ERM inhibition (Fig EV5, B-C), highlighting the specificity of macrophages even among their myeloid counterparts.

This adaptability may include the ability to employ alternative actin-membrane linkers, such as myosin 1a (Mazerik et al., 2014; Tyska et al., 2005), myosin 1b (Diz-Muñoz et al., 2010), myosin 1e (Gupta et al., 2013; Ouderkerk and Krendel, 2014) and myosin 1f (Navinés-Ferrer and Martín, 2020). In accordance with this hypothesis, while we demonstrated that ERMs are not involved in macrophage phagocytosis, recent studies have identified the involvement of myosins 1e and 1f in this process (Barger et al., 2019; Vorselen et al., 2021). “

Reviewer #1 (Significance (Required)):

The manuscript is important on many accounts: The ERM proteins are considered crucial membrane-cytoskeletal linkers in many cellular systems. The study presents a surprising finding that cortical phenomena requiring membrane-cytoskeletal attachment do not essentially need ERM proteins providing a fundamental conceptual advance. The results from this study will also inform both experimental as well as theoretical studies of cortical organization and dynamics in the future. Furthermore, overexpressed mutant forms of ERMs are used as sensors as well as perturbing agents of cortical actin dynamics in many cellular systems. These utilities can now be further substantiated and if required, revised in light of the results from this study.

I am an immune cell biologist specializing in early lymphocyte activation and cytoskeleton dynamics.

We would like to thank the reviewer for pointing out the importance of our work for our understanding of the function of the cellular cortex, and for highlighting the fact that it may lead to a reinterpretation of the results obtained using ERM mutants.

Reviewer #2 (Evidence, reproducibility and clarity (Required)):

Summary

ERM proteins are known to play a central role in linking the cortical actin cytoskeleton with the plasma membrane, which is involved in regulating a diverse range of actin-rich membrane structures. The authors question the role that ERM proteins play in regulating cell shape changes and migration, specifically in macrophages. To test this, they designed an approach to systematically delete each ERM in macrophages - followed by the production of a triple-ERM ko line (tKO) using HoxB8 myeloid progenitor cells. The tKO line was subjected to a series of in vitro and in vivo experiments - all of which involve a series of imaging techniques to monitor membrane dynamics, protein subcellular organisation and cellular behaviours (e.g. rolling fraction, sticking fraction and chemotaxis). Their overall conclusion is that ERM are dispensable for macrophage membrane structures and migration.

General comments.

The experiments are very well executed. The manuscript in short demonstrates that the ERM proteins are dispensable for macrophage migration (both in 2D and 3D contexts), but there is very little beyond this work that points to what they might be doing instead. In this regard, given that the focus is exclusively on macrophage migration, the work comes across as quite specialised.

We thank the reviewer for appreciating the quality of our work.

We respectfully disagree to the assessment of the limited scope of our findings. Given the crucial importance of the migration of macrophages in numerous physiological functions, our findings will have a wide-ranging impact. Furthermore, and as acknowledged by Reviewers 1 and 3, the discovery that ERMs do not play a universal role in cortical mechanics and in cell migration, as hitherto believed, has relevance for a much broader scientific community beyond just those studying macrophages. By introducing a unique research model (a triple KO for ERMs), our work challenges the results of many previous studies that used less specific molecular tools, such as pharmacological inhibitors or mutants of ERMs.

We acknowledge the fact that although our data convincingly demonstrate that the ERM proteins are dispensable for macrophage migration, they do not reveal alternative functions for these proteins. We agree that it could be interesting to search for alternative functions for ERM proteins in macrophages in future studies. However, we believe that such studies are out of the scope of the present manuscript.

The biggest concern I have is with the in vivo part. It should be noted that the work outlined in the manuscript does not actually address diapedesis, which is monitoring transmigration from the blood into tissue. Rolling and sticking do not define diapedesis. The experiments that the authors have conducted may have captured diapedesis events, but that very much depends on the length of time that the IVM was conducted. The authors would need to qualify their claims in this regard. Removing this work altogether would not lessen the impact, given that diapedesis is not shown. The work would therefore be very much in vitro/ex vivo.

We agree with the reviewer that, due to technical limitations, we only measured the rolling and sticking capacity of the +/-ERM cells and did not measure diapedesis directly. Following Reviewer's comments, we have thus modified the text of the manuscript and no longer use the term 'diapedesis' to describe our in vivo intravital imaging studies.

We also clarified the fact that we did not inject differentiated macrophages into the circulation, but macrophage precursors obtained by the treatment of progenitors with a 1 day treatment only (and not a 7 day treatment) with 20 ng/mL M-CSF.

Here, highlighted in yellow, are the changes to the text (in the Introduction, Results, Methods and Legends sections):

Introduction, p3:

“Surprisingly, we found that ERMs are dispensable for macrophages to migrate in diverse contexts, including in vitro 2D migration and 3D invasion of extracellular matrix, ex vivo tissue infiltration through healthy dermis and tumor tissue, and for the in vivo **adhesion of macrophage precursors to an activated endothelium.**”

Results, p6:

“ERM tKo cells without ezrin, radixin, and moesin exhibit no impairment in their ability to adhere to vascular endothelium in vivo and infiltrate the ear derma or fibrosarcoma.

To further investigate the migratory properties of ERM-deficient cells in vivo, we first assessed their ability **to adhere to activated vascular endothelium** into mice bearing a fibrosarcoma (Gui et al., 2018).”

Results, p8:

“Our study uncovered a surprising finding: ezrin, radixin and moesin are dispensable for key aspects of macrophage behavior, including the formation of lamellipodia and filopodia, the dynamics of membrane ruffles and podosomes, migration in vitro (in 2D or 3D matrices) and ex vivo (into dermis or tumor tissues) as well as for the in vivo **adhesion of macrophage precursors to activated vascular endothelium.**”

Methods, p14:

“In vivo analysis of **adhesion to vascular endothelium** with wide-field intravital microscopy”

And

“HoxB8-progenitors were **directed towards monocyte/macrophage differentiation using a 1 day treatment** with 20 ng/mL mouse M-CSF.”

Figure 4 legend, p33:

“Fig. 4: ERM tKO cells have no defect in adhesion to vascular endothelium in vivo and infiltrate tissues explants ex vivo

A. In vivo adhesion to vascular endothelium

Fibrosarcoma cells were injected into the flank of a mice. After a week, tumor was exposed for intravital microscopy, and the femoral artery of recipient mice was catheterized for injection of exogenous cells. Differentially labeled WT and TKO-ERM macrophage precursors were injected in the blood and their behaviour in tumor blood vessels was assessed by real-time imaging. Rolling fractions were quantified as the percentage of rolling cells in the total flux of cells in each blood vessel, and sticking fractions were quantified as the percentage of rolling cells that firmly adhered for a minimum of 30 seconds.”

Specific questions

How sure are the authors that they are capturing these events in cremasteric venules?

As described in the Results and Methods section and now clarified in the result section, these measurements were not captured in cremasteric venules but in fibrosarcoma tumour blood vessels, where we have previously demonstrated strong recruitment of circulating monocytes to infiltrate tumor tissue (Gui et al., Cancer Immunol. Res. 2018).

Is there any sign of cells being trapped in the microcirculation?

The diameter of the tumor blood vessels analysed is consistent with tumor post-capillary venules, and we have not seen cells trapped in these tumor blood vessels.

The reason for injecting macrophages intravenously is not explained.

We injected cells intravenously in order to compare their capacity to adhere to activated tumor blood vessels by intravital microscopy. This is now clarified in the corresponding result section (p6):

“For that purpose, one day differentiated wild-type or ERM-deficient cells were fluorescently labelled with two different cell trackers, mixed in a 1:1 ratio, and co-injected intra-arterially into recipient mice in order to analyse their behaviour in tumor blood vessels by intravital microscopy.”

Are these experiments modelling intravascular (patrolling) macrophages? Monocytes will typically differentiate into macrophages in tissue.

We again apologize for the lack of clarity. In these experiments, we did not inject fully differentiated (seven days) macrophages but progenitors directed towards monocyte/macrophage differentiation using a 1 day treatment with 20 ng/mL mouse M-CSF. We believe that these experiments model the adhesion/recruitment of monocytes by activated vascular endothelium in the tumor microenvironment. This is now clarified in the results and material sections.

The fact that the cells are able to "roll" and "stick" suggests that they have the complimentary cell adhesion molecules, although this is not addressed in the study.

We agree with the reviewer. Our intravital microscopy analyses indicate that the injected cells have the complementary cell adhesion molecules for firm adhesion to activated tumor blood vessels. Importantly, our data clearly demonstrate that the capacity of ERM-tKO cells to bind vascular endothelium in the tumor microenvironment is similar to that of WT cells (Fig. 4A).

Reviewer #2 (Significance (Required)):

The strength of the manuscript is based on the robust in vitro experiments, however such experiments are difficult to address in vivo - mainly because of the issue that macrophages (unless patrolling macrophages) are not a useful model to investigate for ivm experiments.

We thank the reviewer for recognizing the robustness of our *in vitro* experiments. We fully agree with the reviewer that the *in vivo* experiments are more challenging and that the behaviour of monocytes/(patrolling) macrophages is difficult to mimic *in vivo*. However, we believe that our intravital microscopy analyses are important because they demonstrate that ERM-tKO cells retain the capacity to bind firmly (sticking) to activated tumor blood vessels *in vivo*.

This would be of great interest to the macrophage field, which is quite limited in scope. An advancement in the field would be to learn what is taking over the role of ERM in macrophages. As such, this becomes a report with a series of experiments to confirm that ERM are not involved.

Again, we respectfully disagree with the reviewer, as this work goes against the dogma that ERMs are generally the most important mechanical links between the plasma membrane and the cytoskeleton. By clearly establishing that this is not the case in macrophages, cells whose importance for our immunity justifies the importance of their investigation, this study could make it possible to reconsider the functioning of the cellular cortex and the role of ERMs in other cellular systems. The possible specificity of the macrophage cortex is discussed further on p9 of this article.

Reviewer #3 (Evidence, reproducibility and clarity (Required)):

Verdys and colleagues report an elegant study in which the authors describe that ERM proteins are dispensable for migrating monocyte-derived macrophages. The methods are adequate and the results support the conclusions.

We thank the reviewer for these very supportive comments.

Major points:

- Although the authors demonstrate, by multiple methods, the dispensability of ERM proteins in the migration of macrophages derived from monocytes, the role of these proteins must also be evaluated in the phagocytosis process (another relevant functional aspect of macrophages).

This is an excellent suggestion, which should make it possible to clarify the role of ERMs in this important function of macrophages.

ERM proteins have indeed been proposed as important for macrophage phagocytosis. Importantly, their necessity for the early steps of this process is still debated, as conclusions differ depending on the cellular model used and the type of particle to be internalised (Erwig et al., PNAS USA 2006; Di Pietro et al., Sci. Rep. 2017; Gomez and Descoteaux, Biochem. Biophys. Res. Commun. 2018; Mu et al., Nat. Commun. 2018; Okazaki et al., J. Physiol. Sci. 2020). While the implication of ERMs in the early steps of phagocytosis remains controversial, there seems to be a consensus to implicate ezrin and moesin in phagosome maturation (Defacque et al. EMBO 2000 ; Erwig et al., PNAS USA 2006; Marion et al., Traffic 2011; Gomez and Descoteaux, Biochem. Biophys. Res. Commun. 2018).

We addressed the ability of ERM-depleted macrophages to perform phagocytosis. In particular, we quantified the dynamics of phagocytosis of ovalbumin-coated or IgG-opsonized polystyrene beads, which did not reveal any difference between WT and ERM-depleted macrophages (Fig.1).

Figure 1: Phagocytosis by WT and ERM-tKO macrophages. *HoxB8* macrophages cultured on glass coverslips were exposed to IgG-beads (MOI 4:1) or OVA-beads (MOI 24:1). The beads were centrifuged to synchronize phagocytosis and the experiments were carried out for the indicated times (5 to 120 min).

(A) Representative images of fluorescence microscopy of macrophages exposed to OVA-beads for 60 min are shown. Beads that remained outside the cells were distinguished from ingested beads using anti-ovalbumin antibodies and TRITC-coupled secondary antibodies. Beads inside cells are magenta, beads outside cells are green, F-actin is shown in white and nuclei in blue. Scale bars: 50 μ m or 10 μ m for zooms.

(B-E) The percentages of phagocytosing cells (B, D) and percentages of fully internalized beads (C, E) were quantified. Results are expressed as mean \pm SD of at least 2600 cells/time point from 4 independent experiments; and analyzed with two-way Anova followed by Bonferroni's comparison test.

These results are now included in the revised manuscript as Figure EV3.

- How is the activation of key downstream targets of ERM proteins involved in macrophage migration in KO models?

This is a very pertinent question. However, while ERMs have been described as being downstream of several signalling pathways, their own downstream targets are unfortunately poorly documented and, to our knowledge, none are known in macrophages.

In the context of this article, i.e. macrophage migration, we consider that the main target of ERMs is cortical actin, and in this manuscript, we demonstrate that the functional coupling between actin and the plasma membrane is not affected by complete inactivation of the ERMs.

In different cellular contexts, it has been proposed that ERMs regulate PI3K (Gautreau et al. PNAS USA 1999), Ras (Sperka et al. Plos One 2011) or that they are involved in the initiation of protein translation (Briggs et al. Neoplasia 2012), but these results have not yet been confirmed and we believe they are outside the scope of this study.

Reviewer #3 (Significance (Required)):

Advance: The present study fills a gap in the participation of ERM proteins in cell migration. The results obtained on the dispensability of these proteins in macrophage migration can pave avenues for identifying new processes and proteins associated with migration in this context.

Audience: The audience for this study is very broad.

We again thank the reviewer for recognising the importance of this work for the understanding of cell migration.

My expertise: I have expertise in cellular and molecular biology with a focus on processes associated with cancer. Among the numerous research fronts of the group led by me, we recently identified the EZR gene (which encodes the ezrin protein) as a prognostic marker and molecular target in acute leukemias.

Additional requests from EMBOJ scientific advisors:

Due to the rather brief nature of the reports and the concerns indicated by reviewer #2, I [Ieva Gailite] have consulted with two scientific advisors familiar with immune cell migration and actin cortex function, as well as our journal. Both experts found that further insights would be needed for consideration here along some of the avenues outlined below:

- Assessment of the membrane-to-cortex attachment properties in macrophages vs other cell types - are there differences that would explain the lacking role of ERM proteins in this cell type?

Our collaborators, the Diz Muñoz Lab has assessed membrane to cortex attachment (MCA) via atomic force spectroscopy in a variety of cell types including stem, immune and cancer cells. MCA in HoxB8-derived macrophages seems thereby to be indifferent compared to the other cell lines assessed (Fig 3). Thus, the lack of a phenotype in the triple KO macrophages cannot be simply explained by an underlying general mechanical difference of membrane-to-cortex attachment in macrophages. These unpublished data are not included in this article as they are the subject of work still in progress.

Figure for reviewers removed

- Does manipulation of membrane-cortex attachment, e.g., with the artificial linker generated in the Diz Munoz group, alter macrophage migratory properties per se?

Transfecting macrophages derived from primary cells is not an easy task and our various attempts have unfortunately not been successful. As shown in Figure 3, after electroporation of primary monocyte-derived human macrophages, a large majority of cells died and only an extremely small number of cells expressed the linker, but not enough to allow quantitative analyses of cell migration. Our attempts to generate HoxB8 progenitors expressing the IMC-linker under Doxycycline induction were no more successful.

Figure for reviewers removed

- Is ERM protein independence also extendable to other immune cells, e.g., dendritic cells? Differentiation of the triple KO Hoxb8 cells into dendritic cells could allow testing their migratory behaviour.

This is an excellent suggestion. We took advantage of our HoxB8 cells to test the effect of triple ERM KO on the migration of dendritic cells and pre-osteoclasts.

Firstly, we differentiated our progenitors with GM-CSF for 7 days and analysed the status of the non-adherent cell population. Our FACS analyses revealed that we were able to obtain CD11b +, CD11c high, F4/80- and Ly6C- dendritic-like cells (Fig 4).

Figure 4: Differentiation of HoxB8 progenitors in dendritic-like cells. FACS analyses shows that HoxB8 progenitors differentiated with 40 ng/mL GM-CSF are Ly6C-, CD11b +, CD11c high and F4/80 - dendritic-like cells, compared to M-CSF, which differentiates the same progenitors into Ly6C -, CD11b +, CD11c low, and F4/80 + macrophages.

We quantified a 1.8-fold increase in the capacity of ERM tKO dendritic-like cells to migrate through Matrigel, whereas their capacity to migrate through Collagen I was unaffected (Fig 5).

Figure 5: ERM tKO affects the 3D migration through Matrigel of dendritic-like cells.

Percentages of migration of siRNA-treated HMDM inside collagen I and Matrigel are represented as follows: the technical replicates (dot) of 5 independent experiments (highlighted by different grey colors) are represented. The mean (bar) and SD from the 5 independent experiments are shown. Statistical analysis was done on the mean per experiments using a paired two-tailed t-test.

Then, we differentiated our progenitors with 30 ng/mL M-CSF and 50 ng/mL RANKL for 3 days to obtained osteoclast precursors, as previously described (Raynaud-Messina B. et al. PNAS USA 2018). In this case, no significant effect of ERM tKO on pre-osteoclast migration through the two matrices was observed (Fig 6).

Figure 6: ERM tKO does not affect the 3D migration of pre-osteoclasts.

Percentages of migration inside collagen I and Matrigel are represented as follows: the technical replicates (dot) of 4 independent experiments (highlighted by different grey colors) are represented. The mean (bar) and SD from the independent experiments are shown. Statistical analysis was done on the mean per experiments using a paired two-tailed t-test.

These results suggest that the lack of a role for ERMs is specific to the migration of macrophages and pre-osteoclasts and cannot be extrapolated to all leukocytes.

These results are now included and commented in the revised manuscript as Figure EV5.

- Does inducible triple ERM protein knockout or depletion to reduce compensation effects exhibit a similar lack of phenotype?

This is an important issue. Indeed, our results showing no effect on macrophage migration without ERMs indicate either that ERMs are not involved in macrophage migration at all, or that another mechanism takes over after their depletion. It is indeed possible that compensation has taken place during the triple KO of ERMs in our macrophages. To obtain single or triple KO macrophages, we transduced progenitors with lentiviral particles containing one or the three sgRNAs corresponding to each ERM, selected transduced cells with puromycin for 2 days, amplified the progenitors and checked for ERM expression.

Whereas for single KOs, the deletion was successful in the entire cell population (see Fig 7), Western blot analysis of the cell population selected after the triple KO attempt unfortunately revealed that it consisted in a set of cells KO for a single, two or three ERMs:

Figure 7: WB analysis reveals that the whole cell population is not depleted for the three ERMs.

That is why we then carried out a clonal selection to isolate the triple KOs and selected several independent clones which we used for our study. So, between the initial selection with puromycin and obtaining the clones, almost two months passed, giving ample time for compensation to take place. While it is unrealistic to block all compensatory phenomena, particularly those that do not require gene expression regulation, we can limit the time between depletion and observation of the phenotype.

We approached this problem using different experimental strategies.

Firstly, and as suggested by the EMBOJ scientific advisors, we sought to achieve an inducible triple KO of the three ERMs. We used HoxB8-FL progenitors conditionally expressing Cas9, kindly provided by Eva Kiermaier (Limes Institut, Bonn). We first checked that these progenitors could be differentiated into macrophages:

Figure 8: Differentiation of HoxB8-FL progenitors into macrophages. FACS analyses shows that HoxB8-FL progenitors differentiated with 20 ng/mL M-CSF for 7 days are similar to CD11b +, CD11c +, F4/80 + and Ly6C – macrophages from HoxB8 progenitors.

Unfortunately, as in the case of our progenitors constitutively expressing Cas9, we were unable to obtain triple KO without going through a lengthy sub-cloning step.

We therefore tried to prepare new single KO of Moesin, the main ERM in leukocytes, but this time limiting the time after deletion as much as possible. This required carrying out a KO and differentiating macrophages as soon as puromycin selection had been carried out (two days only), without waiting for the cells to be amplified. It was therefore necessary to carry out the KO again before each migration experiment. We performed 5 independent Moesin KOs and verified perfect Moesin depletion in three of them by WB (see example in Fig. 9). Again, analyses revealed no effect on 3D macrophage migration:

Figure 9: Moesin KO in mouse macrophages.

A. Moesin expression level in WT or Moesin KO mouse macrophages treated is representative of 3 independent KO.

E-F. Percentages of migration inside collagen I (E) and Matrigel (F) are represented as follows: the technical replicates (dot) of 4 (collagen I) and 5 (Matrigel) independent experiments (highlighted by different grey colors) are represented. The mean (bar) and SD from the independent experiments are shown. Statistical analysis was done on the mean per experiments using a paired two-tailed t-test.

This result is commented on and included in the revised version of the article (new Fig. EV2).

Finally, we turned to the second suggestion of the EMBOJ scientific advisors, i.e. RNAi simultaneous inhibition of the three ERMs. We had already shown that simple RNAi inhibition of the main ERM, Moesin, in human macrophages did not affect the 3D migration of macrophages in Matrigel or Collagen I. Inhibiting the expression of three different proteins by RNAi in these cells, differentiated from primary monocytes is not often successful, but among 8 trials we succeeded in depleting expression of all three ERMs by $59\% \pm 28\%$, $81\% \pm 10\%$ and $61\% \pm 20\%$ for ezrin, radixin and moesin, respectively in three different donors. As in the case of simple inhibition of Moesin, inhibition of the three ERMs did not affect the 3D migration of macrophages:

Figure 10: ERM depletion in human macrophages by siRNA.

Left: Ezrin, radixin and moesin expression levels of HMDM treated with siCtrl or siRNAs against the three ERMs is representative of 3 independent donors.

Right: Percentages of migration of siRNA-treated HMDM inside collagen I and Matrigel are represented as follows: the technical replicates (dot) of 3 independent experiments (highlighted by different grey colors) are represented. The mean (bar) and SD from the independent experiments are shown. Statistical analysis was done on the mean per experiments using a paired two-tailed t-test.

This result is now included in Figure 1C-E of the paper.

Thus, minimising the time after Moesin KO or RNAi inhibition of the expression of the three ERMs indicates no difference in the migratory phenotype of macrophages, suggesting that if a compensatory mechanism takes over, it is instantaneous and does not require gene expression regulation. We comment on these new results in the discussion of the article.

Dear Renaud,

Thank you for submitting a revised version of your manuscript. I have now looked through your response to the reviewers' comments, and I find it reasonable. Therefore, I would be happy to extend formal acceptance of the manuscript once the following editorial points below are addressed:

1. Please submit up to five keywords.
2. Please remove figures from the manuscript text file.
3. CRediT has replaced the traditional author contributions section because it offers a systematic, machine-readable author contributions format that allows for more effective research assessment. Please remove the Authors Contributions from the manuscript and use the free text boxes beneath each contributing author's name in our online submission system to add specific details on the author's contribution. More information is available in our guide to authors.
4. Please add a "Disclosure and competing interests statement" section after Acknowledgements (further info: <https://www.embopress.org/page/journal/14602075/authorguide#conflictsofinterest>).
5. Please update references according to The EMBO Journal style - where there are more than 10 authors on a paper, the first 10 should be listed, followed by 'et al.' Please also remove DOIs for published papers. Please find further information here: <https://www.embopress.org/page/journal/14602075/authorguide#referencesformat>
6. Please rename the movies into Movie EV1-EV16 and update the callouts accordingly. The legends should be removed from the manuscript text file and zipped with each movie file. Further information is available here: <https://www.embopress.org/page/journal/14602075/authorguide#expandedview>
7. Please compile Appendix figures in one PDF labelled "Appendix" with their legends and add a table of contents including page numbers to the first page.
8. Please add headings for "Figure Legends" and "Expanded View Figure Legends" to manuscript text.
9. In our standard image integrity check, we noted the following issues:
 - The actin blot appears to be re-used three times in the figure 2A. If this is the case, please state so in the figure legends.
 - In figure EV3A, the image appears to be stitched together of several tiles with different contrast settings. There is also a "splice" through the upper part of the tKO1 image. Please clarify how these images were obtained - were they combined from several fields of view? Please also provide source data for this figure.
10. Our data editors have flagged the following issues in figure legends that need correcting:
 - Please provide the legends for figures 3b-c in a sequential manner currently (legend for figure 3c is provided before legend of figure 3b).
 - Please note that the figure 3g is mislabeled as figure 3h in the manuscript.
 - Please define the annotated p values * as well as provide the exact p-values for the same in the legend of figure 1b, g; 2e-f; as appropriate.
 - Please provide the exact p values in the legends of figures EV 3d; EV 5c.
 - Please indicate the statistical test used for data analysis in the legends of figures 2b-g; 3g; 4a; EV 4d.
 - Please add information regarding the number and nature of replicates in the legends of figures 2b-c, g; 3g; 4a; 5d-e, h, j-k.
 - Please define the error bars in the legends of figures 2b-c, g; 3g; 4a; 5d-e, j-k; EV 4d.
 - Please define the scale bar for figure 5i.
 - Please define the black arrowheads in the legend of figure 3e..
 - Please define the red arrows in the legend of figure 5c.
11. Papers published in The EMBO Journal are accompanied online by a 'Synopsis' to enhance discoverability of the manuscript. It consists of A) a short (1-2 sentences) summary of the findings and their significance, B) 3-4 bullet points highlighting key results and C) a synopsis image that is 550x300-600 pixels large (width x height, jpeg or png format). You can either show a model or key data in the synopsis image. Please note that the image size is rather small and that text needs to be readable at the final size. Please send us this information together with the revised manuscript.

With best wishes,

leva

leva Gailite, PhD
Senior Scientific Editor

The EMBO Journal
Meyerhofstrasse 1
D-69117 Heidelberg
Tel: +4962218891309
i.gailite@embojournal.org

We realize that it is difficult to revise to a specific deadline. In the interest of protecting the conceptual advance provided by the work, we recommend a revision within 3 months (25th Aug 2024). Please discuss the revision progress ahead of this time with the editor if you require more time to complete the revisions.

The authors addressed the minor editorial issues.

Dear Renaud,

Thank you for addressing the final editorial points. I am now pleased to inform you that your manuscript has been accepted for publication. Congratulations on a nice study!

Before we forward your manuscript to our publishers, I would like to propose some minor edits in the manuscript abstract and synopsis (please see below and the attached manuscript text file). I have also written a short blurb that will accompany the title of your manuscript in our online system. Please take a look and let me know if any corrections are needed:

Title:

Ezrin, radixin and moesin are dispensable for macrophage migration and cellular cortex mechanics.

Blurb:

Macrophages do not require the ERM family proteins for phagocytosis and cell migration, suggesting divergence of macrophage cortical properties.

Synopsis:

Ezrin, Radixin, and Moesin (collectively known as ERM proteins) serve as crucial cytoskeletal linker proteins connecting the actin cytoskeleton to the plasma membrane. This study shows that a complete loss of ERM proteins in macrophages does not affect the mechanics of their actin cortex and their capacity to migrate.

- Macrophage actin structures are still correctly formed in the absence of ERM proteins.
- Macrophage migration in vitro, ex vivo and in vivo is not affected by ERM depletion.
- The mechanical properties of the macrophage cortex are independent of ERM proteins.

If you have any questions, please do not hesitate to contact the Editorial Office. Thank you for this contribution to The EMBO Journal and congratulations on a successful publication!

With best wishes,

leva

leva Gailite, PhD
Senior Scientific Editor
The EMBO Journal
Meyerhofstrasse 1
D-69117 Heidelberg
Tel: +4962218891309
i.gailite@embojournal.org

Rev_Com_number: RC-2023-02102

New_manu_number: EMBOJ-2023-115975R1

Corr_author: Poincloux

Title: Ezrin, radixin and moesin are dispensable for macrophage migration and cortex mechanics.